# FEDERATED LEARNING WITH QUANTIZED GLOBAL MODEL UPDATES

## ABSTRACT

We study federated learning (FL), which enables mobile devices to utilize their local datasets to collaboratively train a global model with the help of a central server, while keeping data localized. At each iteration, the server broadcasts the current global model to the devices for local training, and aggregates the local model updates from the devices to update the global model. Previous work on the communication efficiency of FL has mainly focused on the aggregation of model updates from the devices, assuming perfect broadcasting of the global model. In this paper, we instead consider broadcasting a compressed version of the global model. This is to further reduce the communication cost of FL, which can be particularly limited when the global model is to be transmitted over a wireless medium. We introduce a lossy FL (LFL) algorithm, in which both the global model and the local model updates are quantized before being transmitted. We analyze the convergence behavior of the proposed LFL algorithm assuming the availability of accurate local model updates at the server. Numerical experiments show that the proposed LFL scheme, which quantizes the global model update (with respect to the global model estimate at the devices) rather than the global model itself, significantly outperforms other existing schemes studying quantization of the global model at the PS-to-device direction. Also, the performance loss of the proposed scheme is marginal compared to the fully lossless approach, where the PS and the devices transmit their messages entirely without any quantization.

## 1 INTRODUCTION

Federated learning (FL) enables wireless devices to collaboratively train a global model by utilizing locally available data and computational capabilities under the coordination of a parameter server (PS) while the data never leaves the devices McMahan & Ramage (2017).

In FL with $M$ devices the goal is to minimize a loss function $F(\boldsymbol{\theta}) = \sum_{m=1}^{M} \frac{B_m}{B} F_m(\boldsymbol{\theta})$ with respect to the global model $\boldsymbol{\theta} \in \mathbb{R}^d$, where $F_m(\boldsymbol{\theta}) = \frac{1}{B_m} \sum_{\boldsymbol{u} \in \mathcal{B}_m} f(\boldsymbol{\theta}, \boldsymbol{u})$ is the loss function at device $m$, with $\mathcal{B}_m$ representing device $m$'s local dataset of size $B_m$, $B \triangleq \sum_{m=1}^{M} B_m$, and $f(\cdot, \cdot)$ is an empirical loss function. Having access to the global model $\boldsymbol{\theta}$, device $m$ utilizes its local dataset and performs multiple iterations of stochastic gradient descent (SGD) in order to minimize the local loss function $F_m(\boldsymbol{\theta})$. It then sends the local model update to the server, which aggregates the local updates from all the devices to update the global model.

FL mainly targets mobile applications at the network edge, and the wireless communication links connecting these devices to the network are typically limited in bandwidth and power, and suffer from various channel impairments such as fading, shadowing, or interference; hence the need to develop an FL framework with limited communication requirements becomes more vital. While communication-efficient FL has been widely studied, prior works mainly focused on the devices-to-PS links, assuming perfect broadcasting of the global model to the devices at each iteration. In this paper, we design an FL algorithm aiming to reduce the cost of both PS-to-device and devices-to-PS communications. To address the importance of quantization at the PS-to-device direction, we highlight that some devices simply may not have the sufficient bandwidth to receive the global model update when the model size is relatively large, particularly in the wireless setting, where the devices are away from the base station. This would result in consistent exclusion of these devices, resulting in significant performance loss. Moreover, the impact of quantization in the device-to-PS direction is less severe due to the impact of averaging local updates at the PS.

**Related work**   There is a fast-growing body of literature on the communication efficiency of FL targeting restricted bandwidth devices. Several studies address this issue by considering communications with rate limitations, and propose different compression and quantization techniques Konecny et al. (2016); McMahan et al. (2017); Konecny & Richtarik (2018); Dowlin et al. (2016); Konecny et al. (2015); Lin et al. (2018b); He et al. (2018); M. M. Amiri & Gündüz (2020), as well as performing local updates to reduce the frequency of communications from the devices to the PS Lin et al. (2018a); Stich (2019). Statistical challenges arise in FL since the data samples may not be independent and identically distributed (iid) across devices. The common sources of the dependence or bias in data distribution are the participating devices being located in a particular geographic region, and/or at a particular time window P. Kairouz et al. (2019). Different approaches have been studied to mitigate the effect of non-iid data in FL McMahan et al. (2017); Hsieh et al. (2019); Li et al. (2020a); Wang et al. (2020); Eichner et al. (2019); Zhao et al. (2018). Also, FL suffers from a significant variability in the system, which is mainly due to the hardware, network connectivity, and available power associated with different devices Li et al. (2019). Active device selection schemes have been introduced to alleviate significant variability in FL systems, where a subset of devices share the resources and participate at each iteration of training Kang et al. (2019); Nishio & Yonetani (2019); Amiri et al. (2020b); Yang et al. (2020; 2019). There have also been efforts in developing convergence guarantees for FL under various scenarios, considering iid data across the devices Stich (2019); Wang & Joshi (2019); Woodworth et al. (2019); Zhou & Cong (2018); Koloskova et al. (2020), non-iid data Koloskova et al. (2020); Li et al. (2020a); Haddadpour & Mahdavi (2019); Li et al. (2020c), participation of all the devices Khaled et al. (2020); Wang et al. (2019); Yu et al. (2018); Huo et al. (2020), or only a subset of devices at each iteration Li et al. (2020b); Karimireddy et al. (2020); Rizk et al. (2020); Li et al. (2020c); Amiri et al. (2020a), and FL under limited communication constraints Amiri et al. (2020a); Recht et al. (2011); Alistarh et al. (2018).

FL with compressed global model transmission has been studied recently in Caldas et al. (2019); Tang et al. (2019) aiming to alleviate the communication footprint from the PS to the devices. The global model parameters are relatively skewed/diverse and the efficiency of quantization diminishes significantly when the peak-to-average ratio of the parameters is large. To overcome this, in Caldas et al. (2019) the PS first employs a linear transform in order to spread the information of the global model vector more evenly among its dimensions, and broadcasts a quantized version of the resultant vector, and the devices apply the inverse linear transform to estimate the global model. We highlight that this approach requires a relatively high computational overhead due to employing the linear transform at the PS and its inverse at the devices, where this overhead grows with the size of the model parameters. Furthermore, the performance evaluation in Caldas et al. (2019) is limited to the experimental results On the other hand, in Tang et al. (2019) the PS broadcasts quantized global model with error accumulation to compensate the quantization error.

**Our contributions**   With the exception of Caldas et al. (2019); Tang et al. (2019), the literature on FL considers perfect broadcasting of the global model from the PS to the devices. With this assumption, no matter what type of local update or device-to-PS communication strategy is used, all the devices are synchronized with the same global model at each iteration. In this paper, we instead consider broadcasting a quantized version of the global model update by the PS, which provides the devices with a lossy estimate of the global model (rather than its accurate estimate) with which to perform local training. This further reduces the communication cost of FL, which can be particularly limited for transmission over a wireless medium while serving a massive number of devices. Also, it is interesting to investigate the impact of various hyperparameters on the performance of FL with lossy broadcasting of the global model since FL involves transmission over wireless networks with limited bandwidth. We introduce a lossy FL (LFL) algorithm, where at each iteration the PS broadcasts a compressed version of the global model update to all the devices through quantization. To be precise, the PS exploits the knowledge of the last global model estimate available at the devices as side information to quantize the global model update. The devices recover an estimate of the current global model by combining the received quantized global model update with their previous estimate, and perform local training using their estimate, and return the local model updates, again employing

quantization. The PS updates the global model after receiving the quantized local model updates from the devices. We provide convergence analysis of the LFL algorithm investigating the impact of lossy broadcasting on the performance of FL. Numerical experiments on the MNIST and CIFAR-10 datasets illustrate the efficiency of the proposed LFL algorithm. We observe that the proposed LFL scheme, which leads to a significant communication cost saving, provides a promising performance with no visible gap to the performance of the fully lossless scenario where the communication from both PS-to-device and device-to-PS directions is assumed to be perfect. Also, it is illustrated that the proposed LFL scheme significantly outperforms the schemes introduced in Caldas et al. (2019) and Tang et al. (2019) considering compression from the PS to devices.

The proposed LFL algorithm differs from the approaches in Caldas et al. (2019); Tang et al. (2019), since we propose broadcasting the global model update, with respect to the previous estimate at the devices, rather than the global model itself. We remark that the global model update has less variability/variance and peak-to-average ratio than the global model (see Figure 2), and hence, for the same communication load, the devices can have a more accurate estimate of the global model. However, this would require all the devices to track the global model at each iteration, even if they do not participate in the learning process by sending their local update. We argue that broadcasting the global model update to the whole set of devices, rather than a randomly chosen subset, would introduce limited additional communication cost as broadcasting is typically more efficient than sending independent information to devices. Moreover, in practice, the subset of participating devices remain the same for a number of iterations, until a device leaves or joins. Our algorithm can easily be adopted to such scenarios by sending the global model, rather than the model update, every time the subset of devices changes. Also, compared to the approach in Caldas et al. (2019), the LFL algorithm requires a significantly smaller computational overhead. Furthermore, unlike Caldas et al. (2019), we provide an in-depth convergence analysis of the proposed LFL algorithm. The advantage of the proposed LFL algorithm over the approaches introduced in Caldas et al. (2019); Tang et al. (2019) is shown numerically, where, despite its significantly smaller communication load, it provides considerably higher accuracy.

**Notation** The set of real numbers is denoted by $\mathbb{R}$. For $x \in \mathbb{R}$, $|x|$ returns the absolute value of $x$. For a vector of real numbers $\boldsymbol{x}$, the largest and the smallest absolute values among all the entries of $\boldsymbol{x}$ are represented by $\max\{|\boldsymbol{x}|\}$ and $\min\{|\boldsymbol{x}|\}$, respectively. For an integer $i$, we let $[i] \triangleq \{1, 2, \ldots, i\}$. The $l_2$-norm of vector $\boldsymbol{x}$ is denoted by $\|\boldsymbol{x}\|_2$.

## 2  Lossy Federated Learning (LFL) Algorithm

We consider a lossy PS-to-device transmission, in which the PS sends a compressed version of the global model to the devices. This reduces the communication cost, and can be particularly beneficial when the PS resources are limited, and/or communication takes place over a constrained bandwidth medium. We denote the estimate of the global model $\boldsymbol{\theta}(t)$ at the devices by $\widehat{\boldsymbol{\theta}}(t)$, where $t$ represents the global iteration count. Having recovered $\widehat{\boldsymbol{\theta}}(t)$, the devices perform a $\tau$-step SGD with respect to their local datasets, and transmit their local model updates to the PS using quantization while accumulating the quantization error.

### 2.1  Global Model Broadcasting

In the proposed LFL algorithm, the PS performs stochastic quantization similarly to the QSGD algorithm introduced in Alistarh et al. (2017) with a slight modification to broadcast the information about the global model to the devices. In particular, at global iteration $t$, the PS aims to broadcast the global model update $\boldsymbol{\theta}(t) - \widehat{\boldsymbol{\theta}}(t-1)$ to the devices. We present the stochastic quantization technique we use, denoted by $Q(\cdot, \cdot)$, in Appendix A.

**Lemma 1.** *For the quantization function $\varphi(x, q)$ and vector $\boldsymbol{Q}(\boldsymbol{x}, q)$ given in (21b) and (22), respectively, we have*

$$\mathbb{E}_\varphi\left[\varphi(x, q)\right] = x, \quad \mathbb{E}_\varphi\left[\varphi^2(x, q)\right] \leq x^2 + 1/(4q^2), \tag{1a}$$

$$\mathbb{E}_\varphi\left[\boldsymbol{Q}(\boldsymbol{x}, q)\right] = \boldsymbol{x}, \quad \mathbb{E}_\varphi\left[\|\boldsymbol{Q}(\boldsymbol{x}, q)\|_2^2\right] \leq \|\boldsymbol{x}\|_2^2 + \varepsilon d \|\boldsymbol{x}\|_2^2/(4q^2), \tag{1b}$$

---

**Algorithm 1** LFL

---

1: **for** $t = 0, \ldots, T-1$ **do**
      • **Global model broadcasting**
2:     PS broadcasts $\boldsymbol{Q}\big(\boldsymbol{\theta}(t) - \widehat{\boldsymbol{\theta}}(t-1), q_1\big)$
3:     $\widehat{\boldsymbol{\theta}}(t) = \widehat{\boldsymbol{\theta}}(t-1) + \boldsymbol{Q}\big(\boldsymbol{\theta}(t) - \widehat{\boldsymbol{\theta}}(t-1), q_1\big)$
      • **Local update aggregation**
4:     **for** $m = 1, \ldots, M$ in parallel **do**
5:        Device $m$ transmits $\boldsymbol{Q}\big(\Delta\boldsymbol{\theta}_m(t) + \boldsymbol{\delta}_m(t), q_2\big) = \boldsymbol{Q}\big(\boldsymbol{\theta}_m^{\tau+1}(t) - \widehat{\boldsymbol{\theta}}(t) + \boldsymbol{\delta}_m(t), q_2\big)$
6:     **end for**
7:     $\boldsymbol{\theta}(t+1) = \widehat{\boldsymbol{\theta}}(t) + \sum_{m=1}^{M} \frac{B_m}{B} \boldsymbol{Q}\big(\Delta\boldsymbol{\theta}_m(t) + \boldsymbol{\delta}_m(t), q_2\big)$
8: **end for**

---

*where $\mathbb{E}_\varphi$ represents expectation with respect to the quatization function $\varphi(\cdot, \cdot)$, and $0 \leq \varepsilon \leq 1$ is defined as $\varepsilon \triangleq (\max\{|\boldsymbol{x}|\} - \min\{|\boldsymbol{x}|\})^2 / \|\boldsymbol{x}\|_2^2$.*

The proof of Lemma 1 is provided in Appendix B. We highlight that the value of $\varepsilon$ depends on the skewness of the magnitudes of the entries of $\boldsymbol{x}$, where it increases for a more skewed entries with a higher variance. We have $\varepsilon = 0$, if and only if all the entries of $\boldsymbol{x}$ have the same magnitude, and $\varepsilon = 1$, if and only if $\boldsymbol{x}$ has only one non-zero entry.

Given a quantization level $q_1$, the PS broadcasts $\boldsymbol{Q}\big(\boldsymbol{\theta}(t) - \widehat{\boldsymbol{\theta}}(t-1), q_1\big)$ to the devices at global iteration $t$. Then the devices obtain the following estimate of $\boldsymbol{\theta}(t)$:

$$\widehat{\boldsymbol{\theta}}(t) = \widehat{\boldsymbol{\theta}}(t-1) + \boldsymbol{Q}\big(\boldsymbol{\theta}(t) - \widehat{\boldsymbol{\theta}}(t-1), q_1\big), \tag{2}$$

which is equivalent to $\widehat{\boldsymbol{\theta}}(t) = \boldsymbol{\theta}(0) + \sum_{i=1}^{t} \boldsymbol{Q}\big(\boldsymbol{\theta}(i) - \widehat{\boldsymbol{\theta}}(i-1), q_1\big)$, where we assumed that $\widehat{\boldsymbol{\theta}}(0) = \boldsymbol{\theta}(0)$. We note that, having the knowledge of the compressed vector $\boldsymbol{Q}\big(\boldsymbol{\theta}(i) - \widehat{\boldsymbol{\theta}}(i-1), q_1\big)$, $\forall i \in [t]$, the PS can also track $\widehat{\boldsymbol{\theta}}(t)$ at each iteration.

### 2.2 Local Update Aggregation

After recovering $\widehat{\boldsymbol{\theta}}(t)$, device $m$ performs a $\tau$-step local SGD, where the $i$-th step corresponds to $\boldsymbol{\theta}_m^{i+1}(t) = \boldsymbol{\theta}_m^i(t) - \eta_m^i(t)\nabla F_m\big(\boldsymbol{\theta}_m^i(t), \xi_m^i(t)\big)$, $i \in [\tau]$, where $\boldsymbol{\theta}_m^1(t) = \widehat{\boldsymbol{\theta}}(t)$, and $\xi_m^i(t)$ denotes the local mini-batch chosen uniformly at random from the local dataset $\mathcal{B}_m$. It then aims to transmit local model update $\Delta\boldsymbol{\theta}_m(t) = \boldsymbol{\theta}_m^{\tau+1}(t) - \widehat{\boldsymbol{\theta}}(t)$ through quantization with error compensation and transmits $\boldsymbol{Q}\big(\Delta\boldsymbol{\theta}_m(t) + \boldsymbol{\delta}_m(t), q_2\big)$ using a quantization level $q_2$, where $\boldsymbol{\delta}_m(t)$ retains the quantization error, and is updated as

$$\boldsymbol{\delta}_m(t+1) = \Delta\boldsymbol{\theta}_m(t) + \boldsymbol{\delta}_m(t) - \boldsymbol{Q}\big(\Delta\boldsymbol{\theta}_m(t) + \boldsymbol{\delta}_m(t), q_2\big), \tag{3}$$

where we set $\boldsymbol{\delta}_m(0) = \boldsymbol{0}$. Having received $\boldsymbol{Q}\big(\Delta\boldsymbol{\theta}_m(t) + \boldsymbol{\delta}_m(t), q_2\big)$ from device $m$, $\forall m \in [M]$, the PS updates the global model as

$$\boldsymbol{\theta}(t+1) = \widehat{\boldsymbol{\theta}}(t) + \sum_{m=1}^{M} \frac{B_m}{B} \boldsymbol{Q}\big(\Delta\boldsymbol{\theta}_m(t) + \boldsymbol{\delta}_m(t), q_2\big). \tag{4}$$

Algorithm 1 summarizes the proposed LFL algorithm.

**Remark 1.** *We do not consider error compensation at the PS with LFL since we have observed performance degradation numerically when compensating the quantization error at the PS. We argue that LFL naturally accumulates the quantization error at the PS since it sends the quatized global model update with respect to the last global model estimate at the devices. We further highlight that the proposed approach is not limited to any specific quantization technique, and any compression technique can be used within the proposed framework.*

## 3 Convergence Analysis of LFL Algorithm

Here we analyze the convergence behaviour of LFL, where for simplicity of the analysis, we assume that the devices can transmit their local updates, $\Delta\boldsymbol{\theta}_m(t)$, $\forall m$, accurately/in a lossless fashion to the PS, and focus on the impact of lossy broadcasting on the convergence.

### 3.1 Preliminaries

We denote the optimal solution minimizing loss function $F(\boldsymbol{\theta})$ by $\boldsymbol{\theta}^*$, and the minimum loss as $F^*$, i.e., $\boldsymbol{\theta}^* \triangleq \arg\min_{\boldsymbol{\theta}} F(\boldsymbol{\theta})$, and $F^* \triangleq F(\boldsymbol{\theta}^*)$. We also denote the minimum value of the local loss function at device $m$ by $F_m^*$. We further define $\Gamma \triangleq F^* - \sum_{m=1}^{M} \frac{B_m}{B} F_m^*$, where $\Gamma \geq 0$, and its magnitude indicates the bias in the data distribution across devices.

For ease of analysis, we set $\eta_m^i(t) = \eta(t)$. Thus, the $i$-th step SGD at device $m$ is given by

$$\boldsymbol{\theta}_m^{i+1}(t) = \boldsymbol{\theta}_m^i(t) - \eta(t)\nabla F_m\left(\boldsymbol{\theta}_m^i(t), \xi_m^i(t)\right), \quad i \in [\tau], m \in [M], \tag{5}$$

where $\boldsymbol{\theta}_m^1(t) = \widehat{\boldsymbol{\theta}}(t)$, given in (2). Device $m$ transmits the local model update

$$\Delta\boldsymbol{\theta}_m(t) = \boldsymbol{\theta}_m^{\tau+1}(t) - \widehat{\boldsymbol{\theta}}(t) = -\eta(t)\sum_{i=1}^{\tau}\nabla F_m\left(\boldsymbol{\theta}_m^i(t), \xi_m^i(t)\right), \quad m \in [M], \tag{6}$$

and the PS updates the global model as

$$\boldsymbol{\theta}(t+1) = \widehat{\boldsymbol{\theta}}(t) - \eta(t)\sum_{m=1}^{M}\sum_{i=1}^{\tau}\frac{B_m}{B}\nabla F_m\left(\boldsymbol{\theta}_m^i(t), \xi_m^i(t)\right). \tag{7}$$

**Assumption 1.** *The expected squared $l_2$-norm of the stochastic gradients are bounded, i.e.,*

$$\mathbb{E}_\xi\left[\left\|\nabla F_m\left(\boldsymbol{\theta}_m^i(t), \xi_m^i(t)\right)\right\|_2^2\right] \leq G^2, \quad \forall i \in [\tau], \forall m \in [M], \forall t. \tag{8}$$

**Assumption 2.** *The loss functions $F_1, \ldots, F_M$ are $L$-smooth; that is, $\forall \boldsymbol{v}, \boldsymbol{w} \in \mathbb{R}^d$,*

$$2\left(F_m(\boldsymbol{v}) - F_m(\boldsymbol{w})\right) \leq 2\langle\boldsymbol{v} - \boldsymbol{w}, \nabla F_m(\boldsymbol{w})\rangle + L\left\|\boldsymbol{v} - \boldsymbol{w}\right\|_2^2, \quad \forall m \in [M]. \tag{9}$$

### 3.2 Strongly Convex Loss Function

Here we provide convergence analysis assuming that the loss functions $F_1, \ldots, F_M$ are $\mu$-strongly convex; that is, $\forall \boldsymbol{v}, \boldsymbol{w} \in \mathbb{R}^d$,

$$2\left(F_m(\boldsymbol{v}) - F_m(\boldsymbol{w})\right) \geq 2\langle\boldsymbol{v} - \boldsymbol{w}, \nabla F_m(\boldsymbol{w})\rangle + \mu\left\|\boldsymbol{v} - \boldsymbol{w}\right\|_2^2, \quad \forall m \in [M]. \tag{10}$$

In the following theorem, whose proof is provided in Appendix C, we present the convergence rate of the LFL algorithm assuming that the devices can send their local updates accurately.

**Theorem 1.** *Let $0 < \eta(t) \leq \min\left\{1, \frac{1}{\mu\tau}\right\}$, $\forall t$. We have*

$$\mathbb{E}\left[\left\|\boldsymbol{\theta}(t) - \boldsymbol{\theta}^*\right\|_2^2\right] \leq \left(\prod_{i=0}^{t-1} A(i)\right)\left\|\boldsymbol{\theta}(0) - \boldsymbol{\theta}^*\right\|_2^2 + \sum_{j=0}^{t-1} B(j)\prod_{i=j+1}^{t-1} A(i), \tag{11a}$$

*where*

$$A(i) \triangleq 1 - \mu\eta(i)\left(\tau - \eta(i)(\tau-1)\right), \tag{11b}$$

$$B(i) \triangleq \left(1 - \mu\eta(i)\left(\tau - \eta(i)(\tau-1)\right)\right)\left(\frac{\eta(i-1)\tau G}{2q_1}\right)^2\varepsilon d + \eta^2(i)(\tau^2 + \tau - 1)G^2$$

$$+ \left(1 + \mu(1 - \eta(i))\right)\eta^2(i)G^2\frac{\tau(\tau-1)(2\tau-1)}{6} + 2\eta(i)(\tau-1)\Gamma, \tag{11c}$$

*for some $0 \leq \varepsilon \leq 1$, and the expectation is with respect to the stochastic gradient function and stochastic quantization.*

**Corollary 1.** *From the $L$-smoothness of the loss function, for $0 < \eta(t) \leq \min\left\{1, \frac{1}{\mu\tau}\right\}$, $\forall t$, and a total of $T$ global iterations, it follows that*

$$\mathbb{E}\left[F(\boldsymbol{\theta}(T))\right] - F^* \leq \frac{L}{2}\mathbb{E}\left[\left\|\boldsymbol{\theta}(T) - \boldsymbol{\theta}^*\right\|_2^2\right]$$

$$\leq \frac{L}{2}\left(\prod_{i=0}^{T-1} A(i)\right)\left\|\boldsymbol{\theta}(0) - \boldsymbol{\theta}^*\right\|_2^2 + \frac{L}{2}\sum_{j=0}^{T-1} B(j)\prod_{i=j+1}^{T-1} A(i), \tag{12}$$

*where the last inequality follows from (11a). Considering $\eta(t) = \eta$ and $\tau = 1$, we have*

$$\mathbb{E}\left[F(\boldsymbol{\theta}(T))\right] - F^* \leq \frac{L}{2}(1 - \mu\eta)^T\left\|\boldsymbol{\theta}(0) - \boldsymbol{\theta}^*\right\|_2^2$$

$$+ \frac{L}{2}\left((1 - \mu\eta)\left(\frac{\varepsilon d}{4q_1^2}\right) + 1\right)\left(1 - (1 - \mu\eta)^T\right)\left(\frac{\eta G^2}{\mu}\right). \tag{13}$$

**Asymptotic convergence analysis** Here we show that, for a decreasing learning rate over time, such that $\lim_{t\to\infty} \eta(t) = 0$, and given small enough $\varepsilon$, $\lim_{T\to\infty} \mathbb{E}\left[F(\boldsymbol{\theta}(T))\right] - F^* = 0$. For $0 < \eta(t) \leq \min\{1, \frac{1}{\mu\tau}\}$, we have $0 \leq A(t) < 1$, and $\lim_{T\to\infty} \prod_{i=0}^{T-1} A(i) = 0$. For simplicity, assume $\eta(t) = \frac{\alpha}{t+\beta}$, for constant values $\alpha$ and $\beta$. For $j \gg 0$, $B(j) \to 0$, and for limited $j$ values, $\prod_{i=j+1}^{T-1} A(i) \to 0$, and so, according to (12), $\lim_{T\to\infty} \mathbb{E}\left[F(\boldsymbol{\theta}(T))\right] - F^* = 0$.

### 3.3 Non-Convex Loss Function

Next, we provide convergence guarantees of the proposed LFL scheme for $L$-smooth and non-convex loss functions $F_1, \ldots, F_M$. For the non-convex case, we provide a weaker notion of convergence Liu & Wright (2015) $\lim_{T\to\infty} \mathbb{E}\left[\|\nabla F(\boldsymbol{\theta}(T))\|_2^2\right] \to 0$. In the following theorem, we bound $\frac{1}{\sum_{t=0}^{T-1}\eta(t)} \sum_{t=0}^{T-1} \eta(t)\mathbb{E}\left[\|\nabla F(\boldsymbol{\theta}(t))\|_2^2\right]$ with the proof provided in Appendix F.

**Theorem 2.** *Performing the LFL algorithm for $T \geq 1$ global iterations assuming that the PS receives the local model updates accurately leads to*

$$\frac{1}{\sum_{t=0}^{T-1}\eta(t)} \sum_{t=0}^{T-1} \eta(t)\mathbb{E}\left[\|\nabla F(\boldsymbol{\theta}(t))\|_2^2\right] \leq \frac{2\left(F(\boldsymbol{\theta}(0)) - F^*\right)}{\tau \sum_{t=0}^{T-1}\eta(t)} + \frac{2\Gamma}{\tau \sum_{t=0}^{T-1}\eta(t)}$$

$$+ \frac{1}{\sum_{t=0}^{T-1}\eta(t)} \sum_{t=0}^{T-1} \left(\frac{\eta(t-1)G}{2q_1}\right)^2 \left(\eta(t)(2\tau-1)L + 2\right)\varepsilon d\tau L$$

$$+ 2G^2\tau L \frac{\sum_{t=0}^{T-1}\eta^2(t)}{\sum_{t=0}^{T-1}\eta(t)} + L^2G^2(\tau-1)(2\tau-1)\frac{\sum_{t=0}^{T-1}\eta^3(t)}{3\sum_{t=0}^{T-1}\eta(t)}. \quad (14)$$

**Choice of $\varepsilon$** We highlight that $\varepsilon$ appears in the convergence analysis of the LFL algorithm in inequalities (45), (63), in which we have

$$\mathbb{E}\left[\left(\max\left\{\left|\sum_{m=1}^{M}\sum_{i=1}^{\tau}\frac{B_m}{B}\nabla F_m\left(\boldsymbol{\theta}_m^i(t-1), \xi_m^i(t-1)\right)\right|\right\}\right.\right.$$

$$\left.\left. - \min\left\{\left|\sum_{m=1}^{M}\sum_{i=1}^{\tau}\frac{B_m}{B}\nabla F_m\left(\boldsymbol{\theta}_m^i(t-1), \xi_m^i(t-1)\right)\right|\right\}\right)^2\right]$$

$$\leq \varepsilon\mathbb{E}\left[\left\|\sum_{m=1}^{M}\sum_{i=1}^{\tau}\frac{B_m}{B}\nabla F_m\left(\boldsymbol{\theta}_m^i(t-1), \xi_m^i(t-1)\right)\right\|_2^2\right], \quad (15)$$

which follows from (26b), where we note that

$$\boldsymbol{\theta}(t) - \widehat{\boldsymbol{\theta}}(t-1) = -\eta(t-1)\sum_{m=1}^{M}\sum_{i=1}^{\tau}\frac{B_m}{B}\nabla F_m\left(\boldsymbol{\theta}_m^i(t-1), \xi_m^i(t-1)\right). \quad (16)$$

On average the entries of $\boldsymbol{\theta}(t) - \widehat{\boldsymbol{\theta}}(t-1)$, given in (16), are not expected to have very diverse magnitudes. Thus, the inequality in (15) should hold for a relatively small value of $\varepsilon$. We have observed numerically that $\varepsilon \approx 10^{-3}$ satisfies inequality (15) for the LFL algorithm.

**Impact of number of local SGD steps $\tau$** For the non-convex case, assuming $\eta(t) = \eta$, $\forall t$, it is easy to verify that the upper bound on $\frac{1}{T}\sum_{t=0}^{T-1}\mathbb{E}\left[\|\nabla F(\boldsymbol{\theta}(t))\|_2^2\right]$, given in Theorem 2, is simplified as follows:

$$h(\tau) = \frac{a_{-1}}{\tau} + a_0 + a_1\tau + a_2\tau^2, \quad (17)$$

where

$$a_{-1} \triangleq \frac{2\left(F(0) - F^* + \Gamma\right)}{\eta T}, \quad a_0 \triangleq \frac{\eta^2 L^2 G^2}{3},$$

$$a_1 \triangleq (2 - \eta L)\eta L G^2\left(1 + \frac{\varepsilon d}{4q_1^2}\right), \quad a_2 \triangleq \eta^2 L^2 G^2\left(\frac{2}{3} + \frac{\varepsilon d}{2q_1^2}\right). \quad (18)$$

We have $dh(\tau)/d\tau = -a_{-1}/\tau^2 + a_1 + 2a_2\tau$, where we note that $a_1 + 2a_2 \geq 0$. For relatively small $a_{-1}$ values, particularly $a_{-1} \leq a_1 + 2a_2$, $h(\tau)$ increases with $\tau$, and $\tau = 1$ minimizes

Table 1: CNN architecture for image classification on MNIST and CIFAR-10.

| MNIST | CIFAR-10 | |
|---|---|---|
| $3 \times 3$ convolutional layer, 32 channels, ReLU activation, same padding | $3 \times 3$ convolutional layer, 32 channels, ReLU activation, same padding | |
| | $3 \times 3$ convolutional layer, 32 channels, ReLU activation, same padding | |
| $2 \times 2$ max pooling | | |
| | dropout with probability 0.2 | |
| $3 \times 3$ convolutional layer, 64 channels, ReLU activation, same padding | $3 \times 3$ convolutional layer, 64 channels, ReLU activation, same padding | |
| | $3 \times 3$ convolutional layer, 64 channels, ReLU activation, same padding | |
| $2 \times 2$ max pooling | | |
| | dropout with probability 0.3 | |
| $3 \times 3$ convolutional layer, 64 channels, ReLU activation, same padding | $3 \times 3$ convolutional layer, 128 channels, ReLU activation, same padding | |
| | $3 \times 3$ convolutional layer, 128 channels, ReLU activation, same padding | |
| $2 \times 2$ max pooling | | |
| fully connected layer with 128 units, ReLU activation | dropout with probability 0.4 | |
| softmax output layer with 10 units | | |

$h(\tau)$; that is, when the training is started close to the optimal solution ($F(0) - F^*$ is relatively small), and/or $\eta$ is relatively large, $\tau = 1$ may be the best choice. On the other hand, for relatively large $a_{-1}$ values, the best $\tau$ can be the nearest integer to the positive solution of $(a_1 + 2a_2\tau)\tau^2 - a_{-1} = 0$.

## 4 NUMERICAL EXPERIMENTS

Here we investigate the performance of the proposed LFL algorithm for image classification on both MNIST LeCun et al. (1998) and CIFAR-10 Krizhevsky & Hinton (2009) datasets utilizing ADAM optimizer Kingma & Ba (2017). We consider $M = 40$ devices, and we measure the performance as the accuracy with respect to the test samples, called *test accuracy*.

**Network architecture** We train different convolutional neural networks (CNNs) with MNIST and CIFAR-10 datasets. The architectures of these CNNs are described in Table 1.

**Data distribution** We consider two data distribution scenarios. In the non-iid scenario, we split the training data samples with the same label (from the same class) to $M/10$ disjoint subsets (assume that $M$ is divisible by 10). We then assign each subset of data samples, selected at random, to a different device. In the iid scenario, we randomly split the training data samples to $M$ disjoint subsets, and assign each subset to a distinct device. We consider non-iid and iid data distributions while training using MNIST and CIFAR-10, respectively.

**State-of-the-art approaches** We consider two approaches with lossy broadcasting introduced in Caldas et al. (2019) and Tang et al. (2019) as the state-of-the-art approaches. With the scheme in Caldas et al. (2019), referred to as lossy transformed global model (LTGM), the PS first employs a linear transform to project the global model. It then quantizes the resultant vector after the linear transform, and sends the quantized vector to the devices. The devices employ the inverse of the linear transform and use the recovered vector for local training. As suggested in Caldas et al. (2019), we consider Walsh-Hadamrd transform and employ the stochastic quantization scheme presented in Appendix A at the PS. On the other hand, with the approach studied in Tang et al. (2019), referred to as lossy global model (LGM), the PS directly quantizes the global model plus the quantization error accumulated from the previous iterations and shares the quantized global model with the devices, while updating the qunatization error. For fairness, we consider the quantization

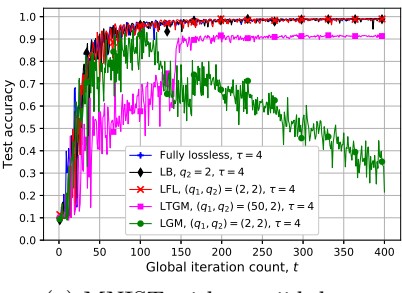 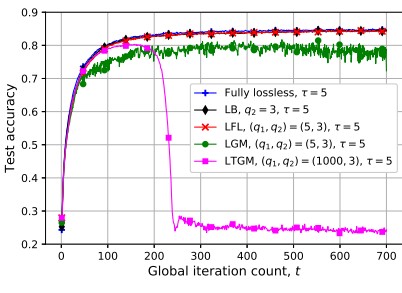

(a) MNIST with non-iid data        (b) CIFAR-10 with iid data

Figure 1: Test accuracy using MNIST and CIFAR-10 for training with local mini-batch size $\left|\xi_m^i(t)\right| = 500$ and $\left|\xi_m^i(t)\right| = 250$, respectively.

scheme presented in Appendix A with the LGM scheme, and assume the same technique for transmission in the device-to-PS direction introduced in Section 2.2.

**Benchmark approaches** We consider the performance of the lossless broadcasting (LB) scenario, where the devices receive the current global model accurately, and perform the quantization with error compensation approach as described in Section 2.2. We highlight that this approach requires transmission of $R_{\mathrm{LB}} = 33d$ bits from the PS, where we assume that each entry of the global model is represented by 33 bits. Thus, the saving ratio in the communication bits of broadcasting from the PS using LFL versus LB is

$$\frac{R_{\mathrm{LB}}}{R_{\mathrm{Q}}} = \frac{33d}{64 + d\left(1 + \log_2(q_1 + 1)\right)} \overset{(a)}{\approx} \frac{33}{1 + \log_2(q_1 + 1)}, \tag{19}$$

where (a) follows assuming that $d \gg 1$. We further consider the performance of the fully lossless approach, where in addition to having the accurate global model at the devices, we assume that the PS receives the local model updates from the devices accurately.

In Figure 1 we illustrate the performance of different approaches for non-iid and iid scenarios using MNIST and CIFAR-10, respectively, for training with $M = 40$ devices. Figure 1a demonstrates test accuracy of different approaches for non-iid data using MNIST with local mini-batch size $\left|\xi_m^i(t)\right| = 500$ and number of local iterations $\tau = 4$. We set $q_2 = 2$ for all the approaches where the devices perform quantization, and $q_1 = 2$ for the LFL and LGM schemes. We observe that the proposed LFL algorithm with $(q_1, q_2) = (2, 2)$ performs as well as the fully lossless and LB approaches, despite a factor of 12.77 savings in the number of bits that need to be broadcast compared to the LB approach. This illustrates the efficiency of the LFL algorithm for the non-iid scenario providing significant communication cost savings without any visible performance degradation. On the other hand, the performance of the LGM algorithm drops after an intermediate number of training iterations, which shows that the quantization level $q_1 = 2$ does not provide the devices with an accurate estimate of the global model to rely on for local training. This is particularly more harmful in later iterations as the algorithm approaches the optimal point where a more accurate estimate of the global model is required for training. We highlight that the proposed LFL algorithm resolves this deficiency with the LGM algorithm through quantizating the global model update rather than the global model providing a more accurate estimate of the global model to the devices even with a relatively small quantization level $q_1 = 2$. Throughout our experiments, we found that the random linear transform with the LTGM scheme is not highly efficient in providing a transformed vector with a relatively small peak-to-average ratio, and the quantization level $q_1$ should be relatively large to guarantee that the algorithm succeeds in learning. Therefore, we set $q_1 = 50$ for the LTGM scheme, which is a relatively large quantization value. The advantage of the proposed LFL algorithm over the LTGM and LGM algorithms for the non-iid scenario can be clearly seen in the figure.

A similar observation is made in Figure 1b illustrating the perforance of different approaches for iid data using CIFAR-10 with local mini-batch size $\left|\xi_m^i(t)\right| = 250$ and number of local iterations $\tau = 5$. The the LFL algorithm with $(q_1, q_2) = (5, 3)$ provides $\times 9.2$ smaller communication load compared to LB with $q_2 = 3$ without any visible performance degradation

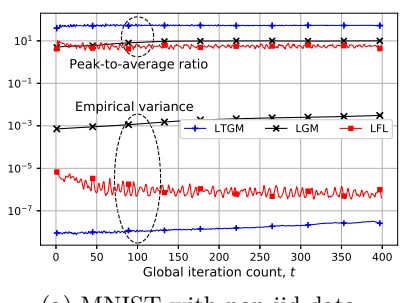
(a) MNIST with non-iid data

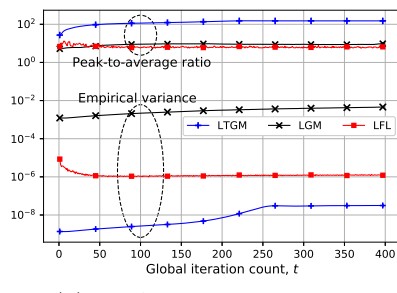
(b) CIFAR-10 with iid data

Figure 2: Empirical variance and peak-to-average ratio of the vector quantized at the PS.

with respect to the fully lossless and LB approaches. It also significantly outperforms the LGM algorithm with $(q_1, q_2) = (5, 3)$, which shows the advantage of quantizing the global model update rather than the global model for iid data. We also observe that the accuracy level of the LTGM algorithm drops significantly after around 200 global iterations even for a large quantization level $q_1 = 1000$, which shows the deficiency of the linear transform to provide a relatively small peak-to-average ratio for the transformed vector.

In Figure 2, we investigate the empirical variance and the peak-to-average ratio of the vector (considering absolute values of its entries) to be quantized at the PS with different schemes for the experimental settings used in Figure 1. This result is provided to better justify the benefits of the proposed LFL scheme over LGM and LTGM shown in Figure 1. We observe that the global model update, which is quantized at the PS with LFL, has significantly smaller empirical variance than the global model, which is quantized at the PS with LGM. This justifies the improvement of LFL over LGM reflecting smaller quantization error when quantizing the global model update rather than the global model, particularly towards the end of training, where the empirical variance of the global model with LGM has an increasing trend over time. Also, both the empirical variance and the peak-to-average ratio of the transformed vector with LTGM increases over time, particularly for training on CIFAR-10. This illustrates that the quantization error increases with time, which may be more harmful towards the end of training while approaching the optimal solution. We note that the relatively small empirical variance of the transformed vector with LTGM is due to the linear transform applied at the PS which scales down the entries of the global model vector. The relatively large peak-to-average ratio indicates that the quantized vector with LTGM may not provide an accurate estimate of the actual transformed vector at the PS.

## 5 CONCLUSION

FL is demanding in terms of bandwidth, particularly when deep networks with huge numbers of parameters are trained across a large number of devices. Communication is typically the major bottleneck, since it involves iterative transmission over a bandwidth-limited wireless medium between the PS and a massive number of devices at the edge. With the goal of reducing the communication cost, we have studied FL with lossy broadcasting, where, in contrast to most of the existing work in the literature, the PS broadcasts a compressed version of the global model to the devices. We have considered broadcasting quantized global model updates from the PS, which can be used to estimate the current global model at the devices for local SGD iterations. The PS aggregates the quantized local model updates from the devices, according to which it updates the global model. We have derived convergence guarantees for the proposed LFL algorithm to analyze the impact of lossy broadcasting on the FL performance assuming accurate local model updates at the PS. Numerical experiments have shown the efficiency of the proposed LFL algorithm in providing an accurate estimate of the global model to the devices, where it performs as well as the fully lossless and LB approaches for both non-iid and iid data despite the significant reduction in the communication load. It also significantly outperforms the LTGM Caldas et al. (2019) and LGM Tang et al. (2019) algorithms studying compression in the PS-to-device direction thanks to quantizing the global model update rather than the global model at the PS.

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

## A  Stochastic quantization

Given $\boldsymbol{x} \in \mathbb{R}^d$, with the $i$-th entry denoted by $x_i$, we define

$$x_{\max} \triangleq \max\{|\boldsymbol{x}|\}, \tag{20a}$$

$$x_{\min} \triangleq \min\{|\boldsymbol{x}|\}. \tag{20b}$$

Given a quantization level $q \geq 1$, we have

$$Q(x_i, q) \triangleq \operatorname{sign}(x_i) \cdot \left( x_{\min} + (x_{\max} - x_{\min}) \cdot \varphi\left( \frac{|x_i| - x_{\min}}{x_{\max} - x_{\min}}, q \right) \right), \quad \text{for } i \in [d], \tag{21a}$$

where $\varphi(\cdot, \cdot)$ is a quantization function defined in the following. For $0 \leq x \leq 1$ and $q \geq 1$, let $l \in \{0, 1, \dots, q-1\}$ be an integer such that $x \in [l/q, (l+1)/q)$. We then define

$$\varphi(x, q) \triangleq \begin{cases} l/q, & \text{with probability } 1 - (xq - l), \\ (l+1)/q, & \text{with probability } xq - l. \end{cases} \tag{21b}$$

We define

$$\boldsymbol{Q}(\boldsymbol{x}, q) \triangleq [Q(x_1, q), \cdots, Q(x_d, q)]^T, \tag{22}$$

and we highlight that it is represented by

$$R_{\mathrm{Q}} = 64 + d\left(1 + \log_2(q+1)\right) \text{ bits}, \tag{23}$$

where 64 bits are used to represent $x_{\max}$ and $x_{\min}$, $d$ bits are used for $\operatorname{sign}(x_i)$, $\forall i \in [d]$, and $d \log_2(q+1)$ bits represent $\varphi\left((|x_i| - x_{\min})/(x_{\max} - x_{\min}), q\right)$, $\forall i \in [d]$. We note that we have modified the QSGD scheme proposed in Alistarh et al. (2017) by normalizing the entries of vector $\boldsymbol{x}$ with $x_{\max} - x_{\min}$ rather than $\|\boldsymbol{x}\|_2$.

## B  Proof of Lemma 1

Given $\varphi(x, q)$ in (21b), we have

$$\mathbb{E}_\varphi[\varphi(x, q)] = \left(\frac{l}{q}\right)(1 + l - xq) + \left(\frac{l+1}{q}\right)(xq - l) = x. \tag{24}$$

Also, we have

$$\mathbb{E}_\varphi\left[\varphi^2(x, q)\right] = \left(\frac{l}{q}\right)^2 (1 + l - xq) + \left(\frac{l+1}{q}\right)^2 (xq - l) = \frac{1}{q^2}\left(-l^2 + 2lxq + xq - l\right)$$

$$= x^2 + \frac{1}{q^2}(xq - l)(1 - xq + l) \overset{\text{(a)}}{\leq} x^2 + \frac{1}{4q^2}, \tag{25}$$

where (a) follows since $(xq - l)(1 - xq + l) \leq 1/4$. According to (24), (25) and the definition of $\boldsymbol{Q}(\boldsymbol{x}, q)$ given in (22), it follows that

$$\mathbb{E}_\varphi[\boldsymbol{Q}(\boldsymbol{x}, q)] = \boldsymbol{x}, \tag{26a}$$

$$\mathbb{E}_\varphi\left[\|\boldsymbol{Q}(\boldsymbol{x}, q)\|_2^2\right] = \sum_{i=1}^d \mathbb{E}_\varphi\left[|Q(x_i, q)|_2^2\right] = (x_{\max} - x_{\min})^2 \sum_{i=1}^d \mathbb{E}_\varphi\left[\varphi^2\left(\frac{|x_i| - x_{\min}}{x_{\max} - x_{\min}}, q\right)\right]$$

$$+ dx_{\min}^2 + 2x_{\min}(x_{\max} - x_{\min}) \sum_{i=1}^d \mathbb{E}_\varphi\left[\varphi\left(\frac{|x_i| - x_{\min}}{x_{\max} - x_{\min}}, q\right)\right]$$

$$\overset{\text{(b)}}{\leq} (x_{\max} - x_{\min})^2 \sum_{i=1}^d \left( \left(\frac{|x_i| - x_{\min}}{x_{\max} - x_{\min}}\right)^2 + \frac{1}{4q^2} \right) + dx_{\min}^2 + 2x_{\min} \sum_{i=1}^d (|x_i| - x_{\min})$$

$$= \|\boldsymbol{x}\|_2^2 + d\frac{(x_{\max} - x_{\min})^2}{4q^2} \overset{\text{(c)}}{\leq} \|\boldsymbol{x}\|_2^2 + \frac{\varepsilon d \|\boldsymbol{x}\|_2^2}{4q^2}, \tag{26b}$$

where (b) follows from (24) and (25), and (c) follows since $\varepsilon = (x_{\max} - x_{\min})^2 / \|\boldsymbol{x}\|_2^2$.

## C  Proof of Theorem 1

We have

$$
\mathbb{E}\left[\|\boldsymbol{\theta}(t+1) - \boldsymbol{\theta}^*\|_2^2\right] = \mathbb{E}\left[\left\|\widehat{\boldsymbol{\theta}}(t) - \boldsymbol{\theta}^*\right\|_2^2\right] + \mathbb{E}\left[\left\|\sum_{m=1}^{M} \frac{B_m}{B}\Delta\boldsymbol{\theta}_m(t)\right\|_2^2\right]
$$
$$
+ 2\mathbb{E}\left[\langle\widehat{\boldsymbol{\theta}}(t) - \boldsymbol{\theta}^*, \sum_{m=1}^{M} \frac{B_m}{B}\Delta\boldsymbol{\theta}_m(t)\rangle\right]. \qquad (27)
$$

In the following, we bound the last two terms on the right hand side (RHS) of (27). From the convexity of $\|\cdot\|_2^2$, it follows that

$$
\mathbb{E}\left[\left\|\sum_{m=1}^{M} \frac{B_m}{B}\Delta\boldsymbol{\theta}_m(t)\right\|_2^2\right] \leq \sum_{m=1}^{M} \frac{B_m}{B}\mathbb{E}\left[\|\Delta\boldsymbol{\theta}_m(t)\|_2^2\right]
$$
$$
= \eta^2(t)\sum_{m=1}^{M}\frac{B_m}{B}\mathbb{E}\left[\left\|\sum_{i=1}^{\tau}\nabla F_m\left(\boldsymbol{\theta}_m^i(t),\xi_m^i(t)\right)\right\|_2^2\right]
$$
$$
\leq \eta^2(t)\tau\sum_{m=1}^{M}\sum_{i=1}^{\tau}\frac{B_m}{B}\mathbb{E}\left[\left\|\nabla F_m\left(\boldsymbol{\theta}_m^i(t),\xi_m^i(t)\right)\right\|_2^2\right] \overset{(a)}{\leq} \eta^2(t)\tau^2 G^2, \qquad (28)
$$

where (a) follows from Assumption 1.

We rewrite the third term on the RHS of (27) as follows:

$$
2\mathbb{E}\left[\langle\widehat{\boldsymbol{\theta}}(t) - \boldsymbol{\theta}^*, \sum_{m=1}^{M}\frac{B_m}{B}\Delta\boldsymbol{\theta}_m(t)\rangle\right]
$$
$$
= 2\eta(t)\sum_{m=1}^{M}\frac{B_m}{B}\mathbb{E}\left[\langle\boldsymbol{\theta}^* - \widehat{\boldsymbol{\theta}}(t), \sum_{i=1}^{\tau}\nabla F_m\left(\boldsymbol{\theta}_m^i(t),\xi_m^i(t)\right)\rangle\right]
$$
$$
= 2\eta(t)\sum_{m=1}^{M}\frac{B_m}{B}\mathbb{E}\left[\langle\boldsymbol{\theta}^* - \widehat{\boldsymbol{\theta}}(t), \nabla F_m\left(\widehat{\boldsymbol{\theta}}(t),\xi_m^1(t)\right)\rangle\right]
$$
$$
+ 2\eta(t)\sum_{m=1}^{M}\frac{B_m}{B}\mathbb{E}\left[\langle\boldsymbol{\theta}^* - \widehat{\boldsymbol{\theta}}(t), \sum_{i=2}^{\tau}\nabla F_m\left(\boldsymbol{\theta}_m^i(t),\xi_m^i(t)\right)\rangle\right]. \qquad (29)
$$

We have

$$
2\eta(t)\sum_{m=1}^{M}\frac{B_m}{B}\mathbb{E}\left[\langle\boldsymbol{\theta}^* - \widehat{\boldsymbol{\theta}}(t), \nabla F_m\left(\widehat{\boldsymbol{\theta}}(t),\xi_m^1(t)\right)\rangle\right]
$$
$$
\overset{(a)}{=} 2\eta(t)\sum_{m=1}^{M}\frac{B_m}{B}\mathbb{E}\left[\langle\boldsymbol{\theta}^* - \widehat{\boldsymbol{\theta}}(t), \nabla F_m\left(\widehat{\boldsymbol{\theta}}(t)\right)\rangle\right]
$$
$$
\overset{(b)}{\leq} 2\eta(t)\sum_{m=1}^{M}\frac{B_m}{B}\mathbb{E}\left[F_m(\boldsymbol{\theta}^*) - F_m\left(\widehat{\boldsymbol{\theta}}(t)\right) - \frac{\mu}{2}\left\|\widehat{\boldsymbol{\theta}}(t) - \boldsymbol{\theta}^*\right\|_2^2\right]
$$
$$
= 2\eta(t)\left(F^* - \mathbb{E}\left[F\left(\widehat{\boldsymbol{\theta}}(t)\right)\right] - \frac{\mu}{2}\mathbb{E}\left[\left\|\widehat{\boldsymbol{\theta}}(t) - \boldsymbol{\theta}^*\right\|_2^2\right]\right)
$$
$$
\overset{(c)}{\leq} -\mu\eta(t)\mathbb{E}\left[\left\|\widehat{\boldsymbol{\theta}}(t) - \boldsymbol{\theta}^*\right\|_2^2\right], \qquad (30)
$$

where (a) follows since $\mathbb{E}_\xi\left[\nabla F_m\left(\boldsymbol{\theta}_m^i(t),\xi_m^i(t)\right)\right] = \nabla F_m\left(\boldsymbol{\theta}_m^i(t)\right), \forall i, m$, (b) is the result of assuming $\mu$-strongly loss functions, and (c) follows since $F^* \leq F\left(\widehat{\boldsymbol{\theta}}(t)\right), \forall t$.

**Lemma 2.** *For $0 < \eta(t) \leq 1$, we have*

$$
2\eta(t)\sum_{m=1}^{M}\frac{B_m}{B}\mathbb{E}\left[\langle\boldsymbol{\theta}^* - \widehat{\boldsymbol{\theta}}(t), \sum_{i=2}^{\tau}\nabla F_m\left(\boldsymbol{\theta}_m^i(t),\xi_m^i(t)\right)\rangle\right]
$$
$$
\leq -\mu\eta(t)(1 - \eta(t))(\tau - 1)\mathbb{E}\left[\left\|\widehat{\boldsymbol{\theta}}(t) - \boldsymbol{\theta}^*\right\|_2^2\right] + \eta^2(t)(\tau - 1)G^2
$$
$$
+ (1 + \mu(1 - \eta(t)))\eta^2(t)G^2\frac{\tau(\tau - 1)(2\tau - 1)}{6} + 2\eta(t)(\tau - 1)\Gamma. \qquad (31)
$$

*Proof.* See Appendix D. $\qquad\square$

By substituting (30) and (31) in (29), it follows that

$$2\mathbb{E}\left[\langle\widehat{\boldsymbol{\theta}}(t)-\boldsymbol{\theta}^*,\sum_{m=1}^{M}\frac{B_m}{B}\Delta\boldsymbol{\theta}_m(t)\rangle\right]$$

$$\leq -\mu\eta(t)(\tau-\eta(t)(\tau-1))\mathbb{E}\left[\left\|\widehat{\boldsymbol{\theta}}(t)-\boldsymbol{\theta}^*\right\|_2^2\right]+\eta^2(t)\,(\tau-1)\,G^2$$

$$+(1+\mu(1-\eta(t)))\eta^2(t)G^2\frac{\tau(\tau-1)(2\tau-1)}{6}+2\eta(t)(\tau-1)\Gamma, \tag{32}$$

which, together with the inequality in (28), leads to the following upper bound on $\mathbb{E}\left[\|\boldsymbol{\theta}(t+1)-\boldsymbol{\theta}^*\|_2^2\right]$, when substituted into (27):

$$\mathbb{E}\left[\|\boldsymbol{\theta}(t+1)-\boldsymbol{\theta}^*\|_2^2\right]\leq(1-\mu\eta(t)(\tau-\eta(t)(\tau-1)))\,\mathbb{E}\left[\left\|\widehat{\boldsymbol{\theta}}(t)-\boldsymbol{\theta}^*\right\|_2^2\right]+\eta^2(t)\,(\tau^2+\tau-1)\,G^2$$

$$+(1+\mu(1-\eta(t)))\eta^2(t)G^2\frac{\tau(\tau-1)(2\tau-1)}{6}+2\eta(t)(\tau-1)\Gamma. \tag{33}$$

**Lemma 3.** *For $\widehat{\boldsymbol{\theta}}(t)$ given in (2), we have*

$$\mathbb{E}\left[\left\|\widehat{\boldsymbol{\theta}}(t)-\boldsymbol{\theta}^*\right\|_2^2\right]\leq\mathbb{E}\left[\|\boldsymbol{\theta}(t)-\boldsymbol{\theta}^*\|_2^2\right]+\left(\frac{\eta(t-1)\tau G}{2q_1}\right)^2\varepsilon d. \tag{34}$$

*for some $0\leq\varepsilon\leq1$.*

*Proof.* See Appendix E. □

According to Lemma 3, the inequality in (34) can be rewritten as follows:

$$\mathbb{E}\left[\|\boldsymbol{\theta}(t+1)-\boldsymbol{\theta}^*\|_2^2\right]\leq(1-\mu\eta(t)(\tau-\eta(t)(\tau-1)))\,\mathbb{E}\left[\|\boldsymbol{\theta}(t)-\boldsymbol{\theta}^*\|_2^2\right]$$

$$+(1-\mu\eta(t)\,(\tau-\eta(t)(\tau-1)))\left(\frac{\eta(t-1)\tau G}{2q_1(t)}\right)^2\varepsilon d+\eta^2(t)\,(\tau^2+\tau-1)\,G^2$$

$$+(1+\mu(1-\eta(t)))\eta^2(t)G^2\frac{\tau(\tau-1)(2\tau-1)}{6}+2\eta(t)(\tau-1)\Gamma. \tag{35}$$

Theorem 1 follows from the inequality in (35) having $0<\eta(t)\leq\min\left\{1,\frac{1}{\mu\tau}\right\},\forall t$.

## D  PROOF OF LEMMA 2

We have

$$2\eta(t)\sum_{m=1}^{M}\frac{B_m}{B}\sum_{i=2}^{\tau}\mathbb{E}\left[\langle\boldsymbol{\theta}^*-\widehat{\boldsymbol{\theta}}(t),\nabla F_m\left(\boldsymbol{\theta}_m^i(t),\xi_m^i(t)\right)\rangle\right]$$

$$=2\eta(t)\sum_{m=1}^{M}\frac{B_m}{B}\sum_{i=2}^{\tau}\mathbb{E}\left[\langle\boldsymbol{\theta}_m^i(t)-\widehat{\boldsymbol{\theta}}(t),\nabla F_m\left(\boldsymbol{\theta}_m^i(t),\xi_m^i(t)\right)\rangle\right]$$

$$+2\eta(t)\sum_{m=1}^{M}\frac{B_m}{B}\sum_{i=2}^{\tau}\mathbb{E}\left[\langle\boldsymbol{\theta}^*-\boldsymbol{\theta}_m^i(t),\nabla F_m\left(\boldsymbol{\theta}_m^i(t),\xi_m^i(t)\right)\rangle\right]. \tag{36}$$

We first bound the first term on the RHS of (36). We have

$$2\eta(t)\sum_{m=1}^{M}\frac{B_m}{B}\sum_{i=2}^{\tau}\mathbb{E}\left[\langle\boldsymbol{\theta}_m^i(t)-\widehat{\boldsymbol{\theta}}(t),\nabla F_m\left(\boldsymbol{\theta}_m^i(t),\xi_m^i(t)\right)\rangle\right]$$

$$\leq\eta(t)\sum_{m=1}^{M}\frac{B_m}{B}\sum_{i=2}^{\tau}\mathbb{E}\left[\frac{1}{\eta(t)}\left\|\boldsymbol{\theta}_m^i(t)-\widehat{\boldsymbol{\theta}}(t)\right\|_2^2+\eta(t)\left\|\nabla F_m\left(\boldsymbol{\theta}_m^i(t),\xi_m^i(t)\right)\right\|_2^2\right]$$

$$\overset{(a)}{\leq}\sum_{m=1}^{M}\frac{B_m}{B}\sum_{i=2}^{\tau}\mathbb{E}\left[\left\|\boldsymbol{\theta}_m^i(t)-\widehat{\boldsymbol{\theta}}(t)\right\|_2^2\right]+\eta^2(t)\,(\tau-1)\,G^2, \tag{37}$$

where (a) follows from Assumption 1. We have

$$
\sum_{m=1}^{M} \frac{B_m}{B} \sum_{i=2}^{\tau} \mathbb{E}\left[\left\|\boldsymbol{\theta}_m^i(t) - \widehat{\boldsymbol{\theta}}(t)\right\|_2^2\right]
$$
$$
= \eta^2(t) \sum_{m=1}^{M} \frac{B_m}{B} \sum_{i=2}^{\tau} \mathbb{E}\left[\left\|\sum_{j=1}^{i} \nabla F_m\left(\boldsymbol{\theta}_m^j(t), \xi_m^j(t)\right)\right\|_2^2\right] \overset{(b)}{\leq} \eta^2(t) G^2 \frac{\tau(\tau-1)(2\tau-1)}{6}, \quad (38)
$$

where (b) follows from the convexity of $\|\cdot\|_2^2$ and Assumption 1. Plugging (38) into (37) yields

$$
2\eta(t) \sum_{m=1}^{M} \frac{B_m}{B} \sum_{i=2}^{\tau} \mathbb{E}\left[\langle \boldsymbol{\theta}_m^i(t) - \widehat{\boldsymbol{\theta}}(t), \nabla F_m\left(\boldsymbol{\theta}_m^i(t), \xi_m^i(t)\right)\rangle\right]
$$
$$
\leq \eta^2(t) G^2 \frac{\tau(\tau-1)(2\tau-1)}{6} + \eta^2(t)(\tau-1) G^2. \quad (39)
$$

For the second term on the RHS of (36), we have

$$
2\eta(t) \sum_{m=1}^{M} \frac{B_m}{B} \sum_{i=2}^{\tau} \mathbb{E}\left[\langle \boldsymbol{\theta}^* - \boldsymbol{\theta}_m^i(t), \nabla F_m\left(\boldsymbol{\theta}_m^i(t), \xi_m^i(t)\right)\rangle\right]
$$
$$
\overset{(a)}{=} 2\eta(t) \sum_{m=1}^{M} \frac{B_m}{B} \sum_{i=2}^{\tau} \mathbb{E}\left[\langle \boldsymbol{\theta}^* - \boldsymbol{\theta}_m^i(t), \nabla F_m\left(\boldsymbol{\theta}_m^i(t)\right)\rangle\right]
$$
$$
\overset{(b)}{\leq} 2\eta(t) \sum_{m=1}^{M} \frac{B_m}{B} \sum_{i=2}^{\tau} \mathbb{E}\left[F_m(\boldsymbol{\theta}^*) - F_m(\boldsymbol{\theta}_m^i(t)) - \frac{\mu}{2}\left\|\boldsymbol{\theta}_m^i(t) - \boldsymbol{\theta}^*\right\|_2^2\right]
$$
$$
= 2\eta(t) \sum_{m=1}^{M} \frac{B_m}{B} \sum_{i=2}^{\tau} \mathbb{E}\left[F_m(\boldsymbol{\theta}^*) - F_m^* + F_m^* - F_m(\boldsymbol{\theta}_m^i(t)) - \frac{\mu}{2}\left\|\boldsymbol{\theta}_m^i(t) - \boldsymbol{\theta}^*\right\|_2^2\right]
$$
$$
= 2\eta(t)(\tau-1)\Gamma + 2\eta(t) \sum_{m=1}^{M} \frac{B_m}{B} \sum_{i=2}^{\tau} \left(F_m^* - \mathbb{E}\left[F_m(\boldsymbol{\theta}_m^i(t))\right]\right)
$$
$$
- \mu\eta(t) \sum_{m=1}^{M} \frac{B_m}{B} \sum_{i=2}^{\tau} \mathbb{E}\left[\left\|\boldsymbol{\theta}_m^i(t) - \boldsymbol{\theta}^*\right\|_2^2\right]
$$
$$
\overset{(c)}{\leq} 2\eta(t)(\tau-1)\Gamma - \mu\eta(t) \sum_{m=1}^{M} \frac{B_m}{B} \sum_{i=2}^{\tau} \mathbb{E}\left[\left\|\boldsymbol{\theta}_m^i(t) - \boldsymbol{\theta}^*\right\|_2^2\right], \quad (40)
$$

where (a) follows since $\mathbb{E}_\xi\left[\nabla F_m\left(\boldsymbol{\theta}(t), \xi_m^i(t)\right)\right] = \nabla F_m\left(\boldsymbol{\theta}(t)\right), \forall i, m, t$; (b) follows from assuming that the loss functions are $\mu$-strongly convex; and (c) follows since $F_m^* \leq F_m(\boldsymbol{\theta}_m^i(t))$, $\forall m$. We have

$$
-\left\|\boldsymbol{\theta}_m^i(t) - \boldsymbol{\theta}^*\right\|_2^2 = -\left\|\boldsymbol{\theta}_m^i(t) - \widehat{\boldsymbol{\theta}}(t)\right\|_2^2 - \left\|\widehat{\boldsymbol{\theta}}(t) - \boldsymbol{\theta}^*\right\|_2^2 - 2\langle \boldsymbol{\theta}_m^i(t) - \widehat{\boldsymbol{\theta}}(t), \widehat{\boldsymbol{\theta}}(t) - \boldsymbol{\theta}^*\rangle
$$
$$
\overset{(a)}{\leq} -\left\|\boldsymbol{\theta}_m^i(t) - \widehat{\boldsymbol{\theta}}(t)\right\|_2^2 - \left\|\widehat{\boldsymbol{\theta}}(t) - \boldsymbol{\theta}^*\right\|_2^2 + \frac{1}{\eta(t)}\left\|\boldsymbol{\theta}_m^i(t) - \widehat{\boldsymbol{\theta}}(t)\right\|_2^2 + \eta(t)\left\|\widehat{\boldsymbol{\theta}}(t) - \boldsymbol{\theta}^*\right\|_2^2
$$
$$
= -(1-\eta(t))\left\|\widehat{\boldsymbol{\theta}}(t) - \boldsymbol{\theta}^*\right\|_2^2 + \left(\frac{1}{\eta(t)} - 1\right)\left\|\boldsymbol{\theta}_m^i(t) - \widehat{\boldsymbol{\theta}}(t)\right\|_2^2, \quad (41)
$$

where (a) follows from Cauchy-Schwarz inequality. Plugging (41) into (40) yields

$$
\frac{2\eta(t)}{M} \sum_{m=1}^{M} \sum_{i=2}^{\tau} \mathbb{E}\left[\langle \boldsymbol{\theta}^* - \boldsymbol{\theta}_m^i(t), \nabla F_m\left(\boldsymbol{\theta}_m^i(t), \xi_m^i(t)\right)\rangle\right]
$$
$$
\leq -\mu\eta(t)(1-\eta(t))(\tau-1)\left\|\widehat{\boldsymbol{\theta}}(t) - \boldsymbol{\theta}^*\right\|_2^2 + \mu(1-\eta(t))\eta^2(t)G^2 \frac{\tau(\tau-1)(2\tau-1)}{6} + 2\eta(t)(\tau-1)\Gamma, \quad (42)
$$

where we used the inequality in (38) and $\eta(t) \leq 1$. Plugging (39) and (42) into (36) completes the proof of Lemma 2.

# E    Proof of Lemma 3

We have

$$\mathbb{E}\left[\left\|\widehat{\boldsymbol{\theta}}(t) - \boldsymbol{\theta}^*\right\|_2^2\right] = \mathbb{E}\left[\left\|\widehat{\boldsymbol{\theta}}(t)\right\|_2^2\right] + \mathbb{E}\left[\|\boldsymbol{\theta}^*\|_2^2\right] - 2\mathbb{E}\left[\langle\widehat{\boldsymbol{\theta}}(t), \boldsymbol{\theta}^*\rangle\right]$$

$$\stackrel{(a)}{=} \mathbb{E}\left[\left\|\widehat{\boldsymbol{\theta}}(t)\right\|_2^2\right] + \mathbb{E}\left[\|\boldsymbol{\theta}^*\|_2^2\right] - 2\mathbb{E}\left[\langle\boldsymbol{\theta}(t), \boldsymbol{\theta}^*\rangle\right], \tag{43}$$

where (a) follows since

$$\mathbb{E}\left[\widehat{\boldsymbol{\theta}}(t)\right] = \mathbb{E}\left[\widehat{\boldsymbol{\theta}}(t-1)\right] + \mathbb{E}\left[\boldsymbol{Q}\big(\boldsymbol{\theta}(t) - \widehat{\boldsymbol{\theta}}(t-1), q_1\big)\right] = \mathbb{E}\left[\boldsymbol{\theta}(t)\right], \tag{44}$$

where the last equality follows from (26a). In the following, we upper bound $\mathbb{E}\left[\left\|\widehat{\boldsymbol{\theta}}(t)\right\|_2^2\right]$.
We have

$$\mathbb{E}\left[\left\|\widehat{\boldsymbol{\theta}}(t)\right\|_2^2\right] = \mathbb{E}\left[\left\|\widehat{\boldsymbol{\theta}}(t-1)\right\|_2^2\right] + \mathbb{E}\left[\left\|\boldsymbol{Q}\big(\boldsymbol{\theta}(t) - \widehat{\boldsymbol{\theta}}(t-1), q_1\big)\right\|_2^2\right]$$

$$+ 2\mathbb{E}\left[\langle\widehat{\boldsymbol{\theta}}(t-1), \boldsymbol{Q}\big(\boldsymbol{\theta}(t) - \widehat{\boldsymbol{\theta}}(t-1), q_1\big)\rangle\right]$$

$$\stackrel{(a)}{\leq} \mathbb{E}\left[\left\|\widehat{\boldsymbol{\theta}}(t-1)\right\|_2^2\right] + \mathbb{E}\left[\left\|\boldsymbol{\theta}(t) - \widehat{\boldsymbol{\theta}}(t-1)\right\|_2^2\right] + \frac{\varepsilon(t)d}{4q_1^2}\mathbb{E}\left[\left\|\boldsymbol{\theta}(t) - \widehat{\boldsymbol{\theta}}(t-1)\right\|_2^2\right]$$

$$+ 2\mathbb{E}\left[\langle\widehat{\boldsymbol{\theta}}(t-1), \boldsymbol{\theta}(t) - \widehat{\boldsymbol{\theta}}(t-1)\rangle\right]$$

$$\stackrel{(b)}{\leq} \mathbb{E}\left[\|\boldsymbol{\theta}(t)\|_2^2\right] + \frac{\varepsilon d}{4q_1^2(t)}\mathbb{E}\left[\left\|\boldsymbol{\theta}(t) - \widehat{\boldsymbol{\theta}}(t-1)\right\|_2^2\right], \tag{45}$$

where (a) follows from (26) for some $0 \leq \varepsilon(t) \leq 1$ defined as

$$\varepsilon(t) \triangleq \frac{\mathbb{E}\left[\left(\max\left\{\left|\boldsymbol{\theta}(t) - \widehat{\boldsymbol{\theta}}(t-1)\right|\right\} - \min\left\{\left|\boldsymbol{\theta}(t) - \widehat{\boldsymbol{\theta}}(t-1)\right|\right\}\right)^2\right]}{\mathbb{E}\left[\left\|\boldsymbol{\theta}(t) - \widehat{\boldsymbol{\theta}}(t-1)\right\|_2^2\right]}, \tag{46}$$

noting that

$$\boldsymbol{\theta}(t) - \widehat{\boldsymbol{\theta}}(t-1) = -\eta(t-1)\sum_{m=1}^M\sum_{i=1}^\tau \frac{B_m}{B}\nabla F_m\big(\boldsymbol{\theta}_m^i(t-1), \xi_m^i(t-1)\big), \tag{47}$$

and in (b) we define $\varepsilon \triangleq \max_t\{\varepsilon(t)\}$. According to (47), from the convexity of $\|\cdot\|_2^2$, it follows
that

$$\mathbb{E}\left[\left\|\boldsymbol{\theta}(t) - \widehat{\boldsymbol{\theta}}(t-1)\right\|_2^2\right] \leq \eta^2(t-1)\sum_{m=1}^M\sum_{i=1}^\tau \frac{B_m}{B}\mathbb{E}\left[\left\|\nabla F_m\big(\boldsymbol{\theta}_m^i(t-1), \xi_m^i(t-1)\big)\right\|_2^2\right]$$

$$\stackrel{(a)}{\leq} \eta^2(t-1)\tau^2 G^2, \tag{48}$$

where (a) follows from Assumption 1. Accordingly, (45) reduces to

$$\mathbb{E}\left[\left\|\widehat{\boldsymbol{\theta}}(t)\right\|_2^2\right] \leq \mathbb{E}\left[\|\boldsymbol{\theta}(t)\|_2^2\right] + \left(\frac{\eta(t-1)\tau G}{2q_1}\right)^2\varepsilon d. \tag{49}$$

Substituting the above inequality into (43) yields

$$\mathbb{E}\left[\left\|\widehat{\boldsymbol{\theta}}(t) - \boldsymbol{\theta}^*\right\|_2^2\right] \leq \mathbb{E}\left[\|\boldsymbol{\theta}(t)\|_2^2\right] + \mathbb{E}\left[\|\boldsymbol{\theta}^*\|_2^2\right] - 2\mathbb{E}\left[\langle\boldsymbol{\theta}(t), \boldsymbol{\theta}^*\rangle\right] + \left(\frac{\eta(t-1)\tau G}{2q_1}\right)^2\varepsilon d$$

$$= \mathbb{E}\left[\|\boldsymbol{\theta}(t) - \boldsymbol{\theta}^*\|_2^2\right] + \left(\frac{\eta(t-1)\tau G}{2q_1}\right)^2\varepsilon d. \tag{50}$$

# F    PROOF OF THEOREM 2

According to the $L$-smoothness of the loss functions $F_1, \ldots, F_m$, we have

$$F(\boldsymbol{\theta}(t+1)) - F(\boldsymbol{\theta}(t)) \le \langle \boldsymbol{\theta}(t+1) - \boldsymbol{\theta}(t), \nabla F(\boldsymbol{\theta}(t)) \rangle + \frac{L}{2} \|\boldsymbol{\theta}(t+1) - \boldsymbol{\theta}(t)\|_2^2. \tag{51}$$

In the following we bound the average of the two terms on the RHS of the above inequality.

**Lemma 4.** *We have*

$$\mathbb{E}\big[\langle \boldsymbol{\theta}(t+1) - \boldsymbol{\theta}(t), \nabla F(\boldsymbol{\theta}(t)) \rangle\big] \le \Big(\frac{\eta(t-1)\tau GL}{2q_1}\Big)^2 \Big(\frac{\varepsilon d\eta(t)(2\tau-1)}{2}\Big)$$

$$+ \eta^3(t) L^2 G^2 \frac{\tau(\tau-1)(2\tau-1)}{6} - \frac{\eta(t)\tau}{2} \mathbb{E}\big[\big\|\nabla F(\boldsymbol{\theta}(t))\big\|_2^2\big]. \tag{52}$$

*Proof.* See Appendix G. $\qquad\square$

**Lemma 5.** *We have*

$$\mathbb{E}\big[\|\boldsymbol{\theta}(t+1) - \boldsymbol{\theta}(t)\|_2^2\big] \le 2\eta^2(t)\tau^2 G^2 + \Big(\frac{\eta(t-1)\tau G}{2q_1}\Big)^2 2\varepsilon d. \tag{53}$$

*Proof.* See Appendix I. $\qquad\square$

Substituting the results in Lemmas 4 and 5 into (51) yields

$$\eta(t)\mathbb{E}\big[\big\|\nabla F(\boldsymbol{\theta}(t))\big\|_2^2\big] \le \frac{2}{\tau}\big(\mathbb{E}\big[F(\boldsymbol{\theta}(t))\big] - \mathbb{E}\big[F(\boldsymbol{\theta}(t+1))\big]\big)$$

$$+ \Big(\frac{\eta(t-1)G}{2q_1}\Big)^2 \big(\eta(t)(2\tau-1)L+2\big)\varepsilon d\tau L + 2\eta^2(t)G^2\tau L + \eta^3(t)L^2 G^2 \frac{(\tau-1)(2\tau-1)}{3}. \tag{54}$$

For any $T$, by summing the above inequality over $t$ we have

$$\sum_{t=0}^{T-1} \eta(t)\mathbb{E}\big[\big\|\nabla F(\boldsymbol{\theta}(t))\big\|_2^2\big] \le \frac{2}{\tau}\big(F(\boldsymbol{\theta}(0)) - \mathbb{E}\big[F(\boldsymbol{\theta}(T))\big]\big)$$

$$+ \sum_{t=0}^{T-1} \Big(\frac{\eta(t-1)G}{2q_1}\Big)^2 \big(\eta(t)(2\tau-1)L+2\big)\varepsilon d\tau L$$

$$+ 2G^2\tau L \sum_{t=0}^{T-1} \eta^2(t) + L^2 G^2 \frac{(\tau-1)(2\tau-1)}{3} \sum_{t=0}^{T-1} \eta^3(t). \tag{55}$$

We bound the first term on the RHS of the above inequality as follows:

$$F(\boldsymbol{\theta}(0)) - \mathbb{E}\big[F(\boldsymbol{\theta}(T))\big] \le F(\boldsymbol{\theta}(0)) - \sum_{m=1}^{M} \frac{B_m}{B} F_m^*$$

$$= F(\boldsymbol{\theta}(0)) - F^* + F^* - \sum_{m=1}^{M} \frac{B_m}{B} F_m^* = F(\boldsymbol{\theta}(0)) - F^* + \Gamma. \tag{56}$$

Substituting the above results in (55) and dividing both sides of the inequality in (55) by $\sum_{t=0}^{T-1} \eta(t)$ complete the proof of Theorem 2.

## G  Proof of Lemma 4

We have

$$\mathbb{E}\big[\langle\boldsymbol{\theta}(t+1)-\boldsymbol{\theta}(t),\nabla F(\boldsymbol{\theta}(t))\rangle\big]=\mathbb{E}\big[\langle\boldsymbol{\theta}(t+1)-\widehat{\boldsymbol{\theta}}(t)-\boldsymbol{\theta}(t)+\widehat{\boldsymbol{\theta}}(t),\nabla F(\boldsymbol{\theta}(t))\rangle\big]$$

$$=\mathbb{E}\big[\langle\boldsymbol{\theta}(t+1)-\widehat{\boldsymbol{\theta}}(t),\nabla F(\boldsymbol{\theta}(t))\rangle\big]-\mathbb{E}\big[\langle\boldsymbol{\theta}(t)-\widehat{\boldsymbol{\theta}}(t),\nabla F(\boldsymbol{\theta}(t))\rangle\big]$$

$$=\mathbb{E}\big[\langle\boldsymbol{\theta}(t+1)-\widehat{\boldsymbol{\theta}}(t),\nabla F(\boldsymbol{\theta}(t))\rangle\big]-\mathbb{E}\big[\langle\boldsymbol{\theta}(t)-\widehat{\boldsymbol{\theta}}(t-1)-\boldsymbol{Q}\big(\boldsymbol{\theta}(t)-\widehat{\boldsymbol{\theta}}(t-1),q_1\big),\nabla F(\boldsymbol{\theta}(t))\rangle\big]$$

$$\overset{\text{(a)}}{=}\mathbb{E}\big[\langle\boldsymbol{\theta}(t+1)-\widehat{\boldsymbol{\theta}}(t),\nabla F(\boldsymbol{\theta}(t))\rangle\big]$$

$$\overset{\text{(b)}}{=}\mathbb{E}\Big[\langle-\eta(t)\sum_{m=1}^{M}\sum_{i=1}^{\tau}\frac{B_m}{B}\nabla F_m\big(\boldsymbol{\theta}_m^i(t)\big),\nabla F(\boldsymbol{\theta}(t))\rangle\Big]$$

$$=-\eta(t)\mathbb{E}\big[\langle\nabla F\big(\widehat{\boldsymbol{\theta}}(t)\big),\nabla F(\boldsymbol{\theta}(t))\rangle\big]-\eta(t)\sum_{i=2}^{\tau}\mathbb{E}\Big[\langle\sum_{m=1}^{M}\frac{B_m}{B}\nabla F_m\big(\boldsymbol{\theta}_m^i(t)\big),\nabla F(\boldsymbol{\theta}(t))\rangle\Big],\quad(57)$$

where (a) follows since $\mathbb{E}_{\varphi}\big[\boldsymbol{Q}\big(\boldsymbol{x},q_1\big)\big]=\boldsymbol{x}$ and the fact that $\boldsymbol{\theta}(t)-\widehat{\boldsymbol{\theta}}(t-1)$ is independent of the stochastic quantization $\boldsymbol{Q}\big(\boldsymbol{\theta}(t)-\widehat{\boldsymbol{\theta}}(t-1),q_1\big)$, and $\mathbb{E}_{\xi}\big[\nabla F_m\big(\boldsymbol{\theta}_m^i(t),\xi_m^i(t)\big)\big]=\nabla F_m\big(\boldsymbol{\theta}_m^i(t)\big),\forall i,m$, results (b). We bound the first term on the RHS of (57) as follows:

$$-\eta(t)\mathbb{E}\big[\langle\nabla F\big(\widehat{\boldsymbol{\theta}}(t)\big),\nabla F(\boldsymbol{\theta}(t))\rangle\big]=\frac{\eta(t)}{2}\Big(\mathbb{E}\big[\|\nabla F(\boldsymbol{\theta}(t))-\nabla F\big(\widehat{\boldsymbol{\theta}}(t)\big)\|_2^2\big]-\mathbb{E}\big[\|\nabla F(\boldsymbol{\theta}(t))\|_2^2\big]$$

$$-\mathbb{E}\big[\|\nabla F\big(\widehat{\boldsymbol{\theta}}(t)\big)\|_2^2\big]\Big)\leq\frac{\eta(t)L^2}{2}\mathbb{E}\big[\|\boldsymbol{\theta}(t)-\widehat{\boldsymbol{\theta}}(t)\|_2^2\big]-\frac{\eta(t)}{2}\mathbb{E}\big[\|\nabla F(\boldsymbol{\theta}(t))\|_2^2\big].\quad(58)$$

**Lemma 6.** *We have*

$$\mathbb{E}\big[\|\boldsymbol{\theta}(t)-\widehat{\boldsymbol{\theta}}(t)\|_2^2\big]\leq\Big(\frac{\eta(t-1)\tau G}{2q_1}\Big)^2\varepsilon d.\quad(59)$$

*Proof.* See Appendix H. □

Plugging (59) into (58) yields

$$-\eta(t)\mathbb{E}\big[\langle\nabla F\big(\widehat{\boldsymbol{\theta}}(t)\big),\nabla F(\boldsymbol{\theta}(t))\rangle\big]\leq\Big(\frac{\eta(t-1)\tau GL}{2q_1}\Big)^2\Big(\frac{\varepsilon d\eta(t)}{2}\Big)-\frac{\eta(t)}{2}\mathbb{E}\big[\|\nabla F(\boldsymbol{\theta}(t))\|_2^2\big].$$
$$(60)$$

The second term on the RHS of (57) is bounded as follows:

$$-\eta(t)\sum_{i=2}^{\tau}\mathbb{E}\Big[\langle\sum_{m=1}^{M}\frac{B_m}{B}\nabla F_m\big(\boldsymbol{\theta}_m^i(t)\big),\nabla F(\boldsymbol{\theta}(t))\rangle\Big]$$

$$=\frac{\eta(t)}{2}\sum_{i=2}^{\tau}\mathbb{E}\Big[\|\sum_{m=1}^{M}\frac{B_m}{B}\nabla F_m\big(\boldsymbol{\theta}_m^i(t)\big)-\nabla F(\boldsymbol{\theta}(t))\|_2^2-\|\sum_{m=1}^{M}\frac{B_m}{B}\nabla F_m\big(\boldsymbol{\theta}_m^i(t)\big)\|_2^2-\|\nabla F(\boldsymbol{\theta}(t))\|_2^2\Big]$$

$$\leq\frac{\eta(t)}{2}\sum_{i=2}^{\tau}\Big(\mathbb{E}\Big[\|\sum_{m=1}^{M}\frac{B_m}{B}\big(\nabla F_m\big(\boldsymbol{\theta}_m^i(t)\big)-\nabla F_m(\boldsymbol{\theta}(t))\big)\|_2^2\Big]-\mathbb{E}\big[\|\nabla F(\boldsymbol{\theta}(t))\|_2^2\big]\Big)$$

$$\overset{\text{(a)}}{\leq}\frac{\eta(t)}{2}\sum_{i=2}^{\tau}\sum_{m=1}^{M}\frac{B_m}{B}\|\big(\nabla F_m\big(\boldsymbol{\theta}_m^i(t)\big)-\nabla F_m(\boldsymbol{\theta}(t))\big)\|_2^2-\frac{\eta(t)(\tau-1)}{2}\mathbb{E}\big[\|\nabla F(\boldsymbol{\theta}(t))\|_2^2\big]$$

$$\leq\frac{\eta(t)L^2}{2}\sum_{i=2}^{\tau}\sum_{m=1}^{M}\frac{B_m}{B}\|\boldsymbol{\theta}_m^i(t)-\boldsymbol{\theta}(t)\|_2^2-\frac{\eta(t)(\tau-1)}{2}\mathbb{E}\big[\|\nabla F(\boldsymbol{\theta}(t))\|_2^2\big]$$

$$\leq\eta(t)L^2\sum_{i=2}^{\tau}\sum_{m=1}^{M}\frac{B_m}{B}\Big(\|\boldsymbol{\theta}_m^i(t)-\widehat{\boldsymbol{\theta}}(t)\|_2^2+\|\boldsymbol{\theta}(t)-\widehat{\boldsymbol{\theta}}(t)\|_2^2\Big)-\frac{\eta(t)(\tau-1)}{2}\mathbb{E}\big[\|\nabla F(\boldsymbol{\theta}(t))\|_2^2\big]$$

$$\overset{\text{(b)}}{\leq}\eta^3(t)L^2G^2\frac{\tau(\tau-1)(2\tau-1)}{6}+\Big(\frac{\eta(t-1)\tau GL}{2q_1}\Big)^2\varepsilon d\eta(t)(\tau-1)-\frac{\eta(t)(\tau-1)}{2}\mathbb{E}\big[\|\nabla F(\boldsymbol{\theta}(t))\|_2^2\big],$$
$$(61)$$

where (a) follows from the convexity of $\|\cdot\|_2^2$, and (b) follows from (38) and (59). Plugging (60) and (61) into (57) yields

$$\mathbb{E}\big[\langle\boldsymbol{\theta}(t+1)-\boldsymbol{\theta}(t),\nabla F(\boldsymbol{\theta}(t))\rangle\big] \leq \Big(\frac{\eta(t-1)\tau GL}{2q_1}\Big)^2\Big(\frac{\varepsilon d\eta(t)(2\tau-1)}{2}\Big)$$
$$+\eta^3(t)L^2G^2\frac{\tau(\tau-1)(2\tau-1)}{6}-\frac{\eta(t)\tau}{2}\mathbb{E}\big[\big\|\nabla F(\boldsymbol{\theta}(t))\big\|_2^2\big]. \quad (62)$$

## H  PROOF OF LEMMA 6

We have

$$\mathbb{E}\Big[\big\|\boldsymbol{\theta}(t)-\widehat{\boldsymbol{\theta}}(t)\big\|_2^2\Big] = \mathbb{E}\Big[\big\|\boldsymbol{\theta}(t)-\widehat{\boldsymbol{\theta}}(t-1)-Q\big(\boldsymbol{\theta}(t)-\widehat{\boldsymbol{\theta}}(t-1),q_1\big)\big\|_2^2\Big]$$
$$= \mathbb{E}\Big[\big\|\boldsymbol{\theta}(t)-\widehat{\boldsymbol{\theta}}(t-1)\big\|_2^2\Big]+\mathbb{E}\Big[\big\|Q\big(\boldsymbol{\theta}(t)-\widehat{\boldsymbol{\theta}}(t-1),q_1\big)\big\|_2^2\Big]$$
$$-2\mathbb{E}\big[\langle\boldsymbol{\theta}(t)-\widehat{\boldsymbol{\theta}}(t-1),Q\big(\boldsymbol{\theta}(t)-\widehat{\boldsymbol{\theta}}(t-1),q_1\big)\rangle\big]$$
$$\overset{(a)}{=} -\mathbb{E}\Big[\big\|\boldsymbol{\theta}(t)-\widehat{\boldsymbol{\theta}}(t-1)\big\|_2^2\Big]+\mathbb{E}\Big[\big\|Q\big(\boldsymbol{\theta}(t)-\widehat{\boldsymbol{\theta}}(t-1),q_1\big)\big\|_2^2\Big]$$
$$\overset{(b)}{\leq} \frac{\varepsilon d}{4q_1^2}\mathbb{E}\Big[\big\|\boldsymbol{\theta}(t)-\widehat{\boldsymbol{\theta}}(t-1)\big\|_2^2\Big]$$
$$= \frac{\varepsilon d}{4q_1^2}\mathbb{E}\Big[\big\|\eta(t-1)\sum_{m=1}^{M}\sum_{i=1}^{\tau}\frac{B_m}{B}\nabla F_m\big(\boldsymbol{\theta}_m^i(t-1),\xi_m^i(t-1)\big)\big\|_2^2\Big]$$
$$\overset{(c)}{\leq} \Big(\frac{\eta(t-1)\tau G}{2q_1}\Big)^2\varepsilon d, \quad (63)$$

where (a) follows since $\boldsymbol{\theta}(t)-\widehat{\boldsymbol{\theta}}(t-1)$ is independent of the stochastic quantization $Q\big(\boldsymbol{\theta}(t)-\widehat{\boldsymbol{\theta}}(t-1),q_1\big)$ and $\mathbb{E}_\varphi\big[Q(\boldsymbol{x},q_1)\big]=\boldsymbol{x}$, the second inequality in (1b) leads to (b), and (c) is the result of the convexity of $\|\cdot\|_2^2$ and Assumption 1.

## I  PROOF OF LEMMA 5

We have

$$\mathbb{E}\Big[\big\|\boldsymbol{\theta}(t+1)-\boldsymbol{\theta}(t)\big\|_2^2\Big] \leq 2\mathbb{E}\Big[\big\|\boldsymbol{\theta}(t+1)-\widehat{\boldsymbol{\theta}}(t)\big\|_2^2\Big]+2\mathbb{E}\Big[\big\|\boldsymbol{\theta}(t)-\widehat{\boldsymbol{\theta}}(t)\big\|_2^2\Big]. \quad (64)$$

For the first term on the RHS of the above inequality, we have

$$2\mathbb{E}\Big[\big\|\boldsymbol{\theta}(t+1)-\widehat{\boldsymbol{\theta}}(t)\big\|_2^2\Big] = 2\mathbb{E}\Big[\big\|\eta(t)\sum_{m=1}^{M}\sum_{i=1}^{\tau}\frac{B_m}{B}\nabla F_m\big(\boldsymbol{\theta}_m^i(t),\xi_m^i(t)\big)\big\|_2^2\Big]$$
$$\overset{(a)}{\leq} 2\eta^2(t)\tau\sum_{m=1}^{M}\sum_{i=1}^{\tau}\frac{B_m}{B}\mathbb{E}\Big[\big\|\nabla F_m\big(\boldsymbol{\theta}_m^i(t),\xi_m^i(t)\big)\big\|_2^2\Big]$$
$$\overset{(b)}{\leq} 2\eta^2(t)\tau^2 G^2, \quad (65)$$

where (a) and (b) follow from the convexity of $\|\cdot\|_2^2$ and Assumption 1, respectively. Plugging (65) and (59) into (64) yields

$$\mathbb{E}\Big[\big\|\boldsymbol{\theta}(t+1)-\boldsymbol{\theta}(t)\big\|_2^2\Big] \leq 2\eta^2(t)\tau^2 G^2+\Big(\frac{\eta(t-1)\tau G}{2q_1}\Big)^2 2\varepsilon d. \quad (66)$$

