# OpenReview forum: "Federated Learning With Quantized Global Model Updates"
_ICLR.cc/2021/Conference — Reject_

### Official Review · AnonReviewer4 · 2020-10-28
**Nice work but concerns about the assumption**

**Rating:** 6
**Confidence:** 3

**Review:**

Summary:
This paper suggests a lossy federated learning (LFL) algorithm where the PS broadcasts a quantized version of the global model to devices. This approach helps reduce the communication cost of federated learning and is in particular useful when the communication bandwidth is limited. Associated convergence analysis is given to present the effect of lossy broadcasting on the performance of federated learning. From experimental results, it is shown that the proposed algorithm with (q_1, q_2) = (2, 2) shows almost the same performance as the lossless schemes while enjoying a considerably reduced amount of communication bits.

Reasons for score:
I vote for accepting. I totally agree with the importance and timeliness of this topic. My main concern is about the justification of the assumption and I hope that the authors can address this in the rebuttal.

Concerns:
- It would be better to explain the practical motivation behind the assumption that the devices transmit the lossless local updates to the PS focusing on lossy broadcasting only. For me, it is not clear how this assumption reflects the practical scenarios.
If the bandwidth of the communication channel between a device and the PS is limited by the amount that only can be achieved by quantization, can't we even do the iterations because the bandwidth required for device-PS communication exceeds the bandwidth limit for this channel?

- For Figure 1, it would be better if some reasons could be presented that the performances of LGM and LTGM tend to change differently over iteration for non-iid and iid data. I think this would help to understand the strength of the proposed method.

---

> ### Author Response · Authors · 2020-11-23
> **Response to Reviewer 4**
>
> First, we thank the reviewer for recognizing the value of our work and also for the valuable comments and suggestions. In this revision, we have tried to address all your comments and suggestions. We hope that the changes in the new submission have addressed your concerns properly. We would also like to highlight that, in the revised submission, we have further provided convergence analysis for smooth and non-convex loss functions, through which we have analyzed the best number of local iterations at the devices.
>
> Q1: It would be better to explain the practical motivation behind the assumption that the devices transmit the lossless local updates to the PS focusing on lossy broadcasting only. For me, it is not clear how this assumption reflects the practical scenarios. If the bandwidth of the communication channel between a device and the PS is limited by the amount that only can be achieved by quantization, can't we even do the iterations because the bandwidth required for device-PS communication exceeds the bandwidth limit for this channel?
>
> A1: We would like to kindly highlight that the assumption of lossless transmission of the local model updates from the devices to the PS is only for simplifying the convergence analysis. We have considered lossy transmission with quantization of the local updates in device-to-PS direction in the system model and the experiments. As also pointed out by the reviewer, we believe that compression in the device-to-PS direction is more important than that in the PS-to-device direction since typically a limited communication bandwidth is shared among the devices in device-to-PS transmission. We have taken this into account throughout the experiments by considering $q_2 \le q_1$. However, to better understand the impact of lossy broadcasting on the convergence result, we have assumed the availablity of the local model updates at the PS accurately. Nevertheless, lossy transmission from the devices to the PS can be incorporated in the convergence analysis of the proposed LFL algorithm by exploiting the analysis established in https://arxiv.org/pdf/1610.02132.pdf, as well as many other papers in the literature considering convergence guarantees with compression at the device-to-PS direction. We believe that this brings more complexity into the convergence analysis, in which case the impact of compression at the PS may not be well understood.
>
> Q2: For Figure 1, it would be better if some reasons could be presented that the performances of LGM and LTGM tend to change differently over iteration for non-iid and iid data. I think this would help to understand the strength of the proposed method.
>
> A2: We highlight that, in the revised submission, we have added Figure 2 illustrating the empirical variance, as well as the peak-to-average ratio of the vector to be quantized at the PS with different schemes. We observe in this figure that the global model update, which is quantized at the PS with the proposed LFL scheme, has significantly smaller empirical variance than the global model, which is quantized at the PS with the LGM scheme. This justifies the improvement of LFL over LGM, particularly towards the end of training, where we see that the empirical variance of the global model with the LGM approach has an increasing trend over time. Furthermore, both the empirical variance and the peak-to-average ratio of the transformed vector with the LTGM scheme increases over time, particularly for training on CIFAR-10. This illustrates that the quantization error increases with time, which may be more harmful towards the end of training while approaching the optimal solution where a more accurate estimate of the global model is required. We note that the relatively small empirical variance of the transformed vector with the LTGM scheme is due to the linear transform applied at the PS which scales down the entries of the global model vector. On the other hand, the relatively large peak-to-average ratio indicates that the quantized vector with the LTGM scheme may not provide an accurate estimate of the actual transformed vector at the PS. We believe that Figure 2 with the above discussion can help better justifying the results observed in Figure 1.

---

### Official Review · AnonReviewer3 · 2020-10-29
**More investigations are required for the applications of the algorithm in practical Federated Learning**

**Rating:** 5
**Confidence:** 3

**Review:**

In the paper, authors propose a novel quantization algorithm to reduce the communication cost between server and clients in federated learning, where the communication are limited and unstable. In their method, both the global model and
the local model updates are quantized before being transmitted. The authors also analyze the convergence of the proposed algorithm under certain conditions and execute federated simulations to evaluate it. Results show that the proposed algorithm converge as fast as the lossless broadcasting with no accuracy degradation.

The following are my concerns:
1) In practice, it is not guaranteed that devices don't change in a certain amount of federated rounds. The proposed algorithm requires that the local devices have a good estimate of the global model, which is not true in real life. In this case, the server needs to send the global model to new devices. Since the target of the paper is to reduce the communication cost, the authors should perform experiments and investigate the effect of new participated devices. What is the upper bound of the new participated devices when this algorithm works better than other compared algorithms?

2) It is better to show the convergence regarding the computational time and communication time.

3) There have multiple tasks more difficult than EMNIST and CIFAR10 in Tensorflow Federated, e.g. Stackoverflow.  MNIST and CIFAR10 is a little bit weak to prove the effectiveness of the proposed algorithm.

---

> ### Author Response · Authors · 2020-11-23
> **Response to Reviewer 3**
>
> Q1: In practice, it is not guaranteed that devices don't change in a certain amount of federated rounds. The proposed algorithm requires that the local devices have a good estimate of the global model, which is not true in real life. In this case, the server needs to send the global model to new devices. Since the target of the paper is to reduce the communication cost, the authors should perform experiments and investigate the effect of new participated devices. What is the upper bound of the new participated devices when this algorithm works better than other compared algorithms?
>
> A1: As pointed out by the reviewer and highlighted in the submitted paper, with the proposed LFL scheme the devices need to know the last global model estimate. Therefore, the PS shares $\widehat{\boldsymbol{\theta}} (t)$ with the new devices joining the training at iteration $t$, and this requires transmission of $33d$ bits assuming 32-bit floating point representation. Per reviewer's suggestion, we are currently conducting experiments allowing new devices to join the training at random considering a joining probability, and compare the performance with other schemes for the same average communication load in the PS-to-device direction. Our goal is to find the joining probability below which the proposed LFL algorithm outperforms the other schemes under consideration.
>
> Q2: It is better to show the convergence regarding the computational time and communication time.
>
> A2: We agree with the reviewer that it is important to investigate the training time in federated learning. However, we would like to remark that the focus of our work is on reducing the communication requirements from the PS to the devices while having a reasonable computational overhead at the devices. We have provided discussions in the submitted paper addressing the training delay. We note that the communication delay is captured with the communication load in both the PS-to-device and device-to-PS directions. On the other hand, as emphasized in the submitted paper, the computation delay of the LTGM scheme is significantly higher than that of both the LFL and LGM schemes since it requires a linear transform and its inverse at the PS and devices, respectively. Whereas, the difference in computation delay between the LFL and LGM schemes is marginal since they both have relatively small computation overhead. As a result, assuming the same communication load which results in the same communication delay, we expect that LFL and LGM have significantly smaller computation delay than LTGM.
>
> Q3: There have multiple tasks more difficult than EMNIST and CIFAR10 in Tensorflow Federated, e.g. Stackoverflow. MNIST and CIFAR10 is a little bit weak to prove the effectiveness of the proposed algorithm.
>
> A3: We are currently working towards executing new experiments considering more complicated datasets, such as CIFAR-100.

---

### Official Review · AnonReviewer2 · 2020-10-29
**Good but not good enough**

**Rating:** 5
**Confidence:** 3

**Review:**

In the setting of Federated Learning, the authors propose to quantize both (1) the model send from PS to devices, and (2) the update from device to PS. Although the idea of model-broadcast-compression has appeared in previous work, the authors improve upon previous works in that (1) the authors propose to compress-and-send model updates instead of the model itself, which have smaller variance and peak-to-average ratio and therefore have more effective quantization compression, and (2) do not require pre-processing like linear transformation. The authors further show the effectiveness of their proposed algorithm, LFL, by offering a convergence analysis under appropriate conditions, and offering thorough experimental evaluations.

Pros:
+ The authors show convergence results under the strong-convex and smooth condition, making this work more theoretically-grounded than previous related work.


Cons:
+ I am not convinced that sending updates instead of sending models is a valuable idea.
Sending updates instead of sending models is definitely not a new idea in the quantization optimization community, but mostly from device to PS direction, and not the other way around. I believe there are some challenges or tradeoffs to do the PS-to-device directions, but the authors did not sufficiently address this in the paper. For example, two trade-offs that I would expect are, (1) now devices have more computation responsibility, but they are usually weak in computation and sensitive in power consumption, (2) we are essentially quantizing the updates twice (once from devices to PS, and once from PS to devices). This double quantization will inevitably lead to larger quantization variance, and therefore slower convergence of the model, especially when we are close to the end of the learning.


+ The contribution is not clearly highlighted in the theory nor in the simulation.
The authors mention that "in Tang et al. (2019) the PS broadcasts quantized global model ... Instead, we propose broadcasting the global model update ... We remark that the global model update has less variability/variance and peak-to-average ratio than the global model..."
The authors indicate that, claimed improvements come from the fact that the model update has a smaller variance/peak-to-average ratio than the model.
I believe that this intuition is important in terms of differentiating this work among all model-broadcast-compression works.
Hence I would expect the authors to conduct corresponding experiments, and corresponding highlights in the analysis, showing that the variance/peak-to-average ratio of model updates are indeed $c$ times smaller than the model itself, and the gain (e.g. reduction of communication cost, reduction of computation iteration) is proportional to $c$.

+ The authors mention that "(some devices might not be able to participate all learning iterates) ... Our algorithm can easily be adapted to such scenarios by sending the global model, rather than the model update".
I would recommend the authors conduct another set of experiments, maybe in the future, to take random devices join/drop-out into consideration.
There should be a join/drop-out probability thresholding, below which LFL is more efficient than the previous work, and above which LFL is less effective.
This would complete the picture of the 'scope' of LFL.

+ The contribution section is way too lengthy. Probably move the second paragraph to the related work section.

+ What is the architecture the authors used to conduct the experiments? And how the communication cost is evaluated?

---

> ### Author Response · Authors · 2020-11-23
> **Response to Reviewer 2 [Questions 3-5]**
>
> Q3: The authors mention that "(some devices might not be able to participate all learning iterates) ... Our algorithm can easily be adapted to such scenarios by sending the global model, rather than the model update". I would recommend the authors conduct another set of experiments, maybe in the future, to take random devices join/drop-out into consideration. There should be a join/drop-out probability thresholding, below which LFL is more efficient than the previous work, and above which LFL is less effective. This would complete the picture of the 'scope' of LFL.
>
> A3: As pointed out by the reviewer, with the proposed LFL scheme the participating devices need to know the last global model estimate. Therefore, the PS shares $\widehat{\boldsymbol{\theta}} (t)$ with the new devices joining the training at iteration $t$, and this requires transmission of $33d$ bits assuming 32-bit floating point representation. Per reviewer's suggestion, we are currently conducting experiments allowing new devices to join the training at random, and compare the performance with other schemes for the same average communication load in the PS-to-device direction. As pointed out by the reviewer, we expect to find a joining probability below which the proposed LFL scheme outperforms the other schemes under consideration.
>
> Q4: The contribution section is way too lengthy. Probably move the second paragraph to the related work section.
>
> A4: We have shortened the contribution section by moving some parts of the second paragraph to the related work section.
>
> Q5: What is the architecture the authors used to conduct the experiments? And how the communication cost is evaluated?
>
> A5: The architectures of the convolutional neural networks (CNNs) used for training on MNIST and CIFAR-10 are described in Table 1 in the submitted paper. Also, communication cost is captured with the number of bits transmitted by the PS. As given in equation (21) in the new submission, with the LFL algorithm, the PS requires to transmit $R_{\rm{Q}} = 64 + d \left( 1 + \log_2(q+1) \right)$ bits, where $64$ bits are used to represent $x_{\rm{max}}$ and $x_{\rm{min}}$, $d$ bits are used for ${\rm{sign}} (x_i)$, $\forall i \in [d]$, and $d \log_2(q+1)$ bits represent $q+1$ quantization levels indicated by $\varphi \left( (\left| x_i \right|-x_{\rm{min}})/ (x_{\rm{max}} - x_{\rm{min}}), q \right)$, $\forall i \in [d]$. Also, we need $33d$ bits to transmit the global model accurately, where each entry of the global model is represented by 33 bits (32 bits and 1 bit are used to represent the magnitude and the sign, respectively).

---

> ### Author Response · Authors · 2020-11-23
> **(Continued) Response to Reviewer 2 [Questions 1-2]**
>
> Thank you for recognizing the value of our work and also for the constructive suggestions. We would like to highlight that, in the revised submission, we have further provided convergence analysis for smooth and non-convex loss functions, through which we have analyzed the best number of local iterations at the devices.
>
> Q1: I am not convinced that sending updates instead of sending models is a valuable idea. Sending updates instead of sending models is definitely not a new idea in the quantization optimization community, but mostly from device to PS direction, and not the other way around. I believe there are some challenges or tradeoffs to do the PS-to-device directions, but the authors did not sufficiently address this in the paper. For example, two trade-offs that I would expect are, (1) now devices have more computation responsibility, but they are usually weak in computation and sensitive in power consumption, (2) we are essentially quantizing the updates twice (once from devices to PS, and once from PS to devices). This double quantization will inevitably lead to larger quantization variance, and therefore slower convergence of the model, especially when we are close to the end of the learning.
>
> A1: We agree with the reviewer that sending the updates has been studied in the distributed/federated learning literature. However, to the best of our knowledge, this is the first work to introduce broadcasting the updates from the PS to the devices. We emphasize that this is motivated since the global model update has smaller empirical variance and peak-to-average ratio than the global model itself resulting in a smaller quantization error for the same quantization level. We have illustrated this in Figure 2 in the revised submission, which compares the empirical variance, as well as the peak-to-average ratio of the vector to be quantized at the PS with different schemes. The efficiency of the proposed LFL scheme has been illustrated in Figure 1, where it performs as well as the fully lossless approach with no compression at the PS and the devices for both MNIST and CIFAR-10 datasets despite significant communication savings in both the PS-to-device and device-to-PS directions.
>
> The additional overhead introduced by the proposed LFL scheme at the devices is limited to adding the received global model update with the last global model estimate available at the devices. We have included more discussions about the computational overhead and the challenges with the proposed LFL algorithm in the contributions section of the revised submission.
>
> Q2: The contribution is not clearly highlighted in the theory nor in the simulation. The authors mention that "in Tang et al. (2019) the PS broadcasts quantized global model ... Instead, we propose broadcasting the global model update ... We remark that the global model update has less variability/variance and peak-to-average ratio than the global model..." The authors indicate that, claimed improvements come from the fact that the model update has a smaller variance/peak-to-average ratio than the model. I believe that this intuition is important in terms of differentiating this work among all model-broadcast-compression works. Hence I would expect the authors to conduct corresponding experiments, and corresponding highlights in the analysis, showing that the variance/peak-to-average ratio of model updates are indeed $c$ times smaller than the model itself, and the gain (e.g. reduction of communication cost, reduction of computation iteration) is proportional to $c$.
>
> A2: As suggested by the reviewer, in the revised submission, we have added Figure 2 illustrating the empirical variance, as well as the peak-to-average ratio of the vector to be quantized at the PS with different schemes. We observe in this figure that the global model update, which is quantized at the PS with the proposed LFL scheme, has significantly smaller empirical variance than the global model, which is quantized at the PS with the LGM scheme. This justifies the improvement of LFL over LGM, particularly towards the end of training, where we see that the empirical variance of the global model with the LGM approach has an increasing trend over time. Also, we would like to highlight that this empirical variance reduction is captured with  in the analytical results, where, as noted in equation (15), we expect  to be a relatively small value which is due to sending the global model update at the PS.

---

### Official Review · AnonReviewer1 · 2020-11-01
**Comments for Federated Learning with Quantized Global Model Updates**

**Rating:** 5
**Confidence:** 4

**Review:**

This paper studies federated learning with quantization. The problem setting is very standard, including both iid and non-iid cases. This work proposes a new algorithm, called lossy FL, to save the communication costs, especially from the broadcasting direction. To my understanding, the algorithm is new but still very similar to double-squeeze (Tang 2019). The presentation of the paper is generally good. However, there are several major issues regarding the quality and significance of this work.

1. The convergence analysis is based on the assumption that the problem is strongly convex. But from the motivation of this work and numerical results, the problems would be nonconvex. The current analysis is restrictive.

2. There is no analytic result of quantifying the upper bound of \tau, which is one of the key differences between federated learning and distributed training. There are already many works there.

3. The comparison with double-squeeze is definitely not fair, since the aggregation rule of the double-squeeze did not consider the \tau step local update.

4. The numerical results are very limited, where the CNN network is very small. There is no need to perform quantization for this neural net. Also, MNIST and CIFAR-10 are all small datasets. I don’t know why there were 40 devices used. To show the advantage of federated learning, the speed up in terms of the training time should be compared and plotted.

---

> ### Author Response · Authors · 2020-11-23
> **Response to Reviewer 1 [Question 4]**
>
> Q4: The numerical results are very limited, where the CNN network is very small. There is no need to perform quantization for this neural net. Also, MNIST and CIFAR-10 are all small datasets. I don’t know why there were 40 devices used. To show the advantage of federated learning, the speed up in terms of the training time should be compared and plotted.
>
> A4: Our goal here is to illustrate the benefits of the proposed quantization approach with respect to the literature. We also would like to point out that the DoubleSqueeze paper also uses the CIFAR-10 dataset. Similarly, many other papers in the literature studying compression in distributed/federated learning, including https://arxiv.org/pdf/1812.07210.pdf, https://arxiv.org/pdf/1602.05629.pdf, https://arxiv.org/pdf/1909.13014.pdf, https://arxiv.org/pdf/2003.10422.pdf, have used CNNs for their experiments with similar architectures, and not much larger number of parameters, compared to the CNN we used. Indeed, we expect that a more complicated CNN with more model parameters will better show the effectiveness of the proposed algorithm. We are currently conducting experiments for CIFAR-100. We further highlight that the proposed scheme is not limited to any number of devices, and we selected 40 devices following the literature https://arxiv.org/pdf/1806.00582.pdf, https://arxiv.org/pdf/2002.08782.pdf, https://arxiv.org/pdf/2003.10422.pdf, https://arxiv.org/pdf/1910.00189.pdf.
>
> We agree with the reviewer that it is important to investigate the training time in federated learning. We point out that, although the focus of our work is on reducing the communication requirements from the PS to the devices while having a reasonable computational overhead at the devices, which naturally have low computational resources, we have provided discussions in the submitted paper addressing the training delay. We note that the communication delay is captured with the communication load in both PS-to-device and device-to-PS directions. On the other hand, as emphasized on the submitted paper, the computation delay of the LTGM scheme is significantly higher than that of both LFL and LGM schemes since it requires linear transform and its inverse at the PS and the devices, respectively. The difference in the computation delay between LFL and LGM schemes is marginal since they both have relatively small computational overhead. As a result, assuming the same communication load which results in the same communication delay, we expect that LFL and LGM have significantly smaller computation delay than LTGM.

---

> ### Author Response · Authors · 2020-11-23
> **(Continued) Response to Reviewer 1 [Questions 1-3]**
>
> First, we would like to highlight that the main difference between the proposed LFL approach and the double-squeeze approach, referred to as LGM in our submission, lies in quantizing the global model updates at the PS rather than the global model. This is motivated by the fact that the global model update has less empirical variance and peak-to-average ratio than the global model resulting in a smaller quantization error for the same quantization level. Hence, for the same communication load, the devices can have a more accurate estimate of the global model using their knowledge of the last global model estimate. In the revised submission, we have added Figure 2 illustrating the empirical variance, as well as the peak-to-average ratio of the vector to be quantized at the PS with different schemes. We observe in this figure that the global model update, which is quantized at the PS with the proposed LFL scheme, has significantly smaller empirical variance than the global model, which is quantized at the PS with the LGM scheme.
>
> Q1:
> The convergence analysis is based on the assumption that the problem is strongly convex. But from the motivation of this work and numerical results, the problems would be nonconvex. The current analysis is restrictive.
>
> A1:
> As suggested by the reviewer, in the revised version of the paper, we have provided convergence analysis for smooth and non-convex loss functions. The convergence rate has been presented in Theorem 2 with the proof provided in Appendix F.
>
> Q2:
> There is no analytic result of quantifying the upper bound of $\tau$, which is one of the key differences between federated learning and distributed training. There are already many works there.
>
> A2:
> In the revised submission, we have analyzed the best number of local iterations $\tau$ after Theorem 2 according to the convergence rate for the non-convex case.
>
> Q3:
> The comparison with double-squeeze is definitely not fair, since the aggregation rule of the double-squeeze did not consider the $\tau$ step local update.
>
> A3:
> We agree with the reviewer that the double-squeeze approach, referred to as LGM in our submission, does not consider multiple local iterations. However, we have adopted this into the federated learning framework by allowing the devices to perform an arbitrary number of local updates. Our goal is to compare the proposed LFL scheme, which introduces quantizing the global model updates at the PS, with that of compressing the global model itself. We would like to emphasize that the improvement of LFL over LGM holds also for the case of a single local iteration at the devices, i.e., $\tau=1$. We did not present this result in the submission since we prefer to keep the system model and the experimental setting more general and in accordance with the standard federated learning setting by allowing devices to perform multiple local iterations.

---

### Author Response · Authors · 2020-11-23
**Initial General Response**

We would like to thank the reviewers for their careful reading of our paper and for their constructive suggestions. Following the reviews, we have updated the paper, and we have tried to address all reviewers' comments and suggestions. We hope that the changes in the new paper have addressed the reviewers' concerns properly. We would like to highlight that, in the revised paper, we have provided convergence analysis for smooth and non-convex loss functions, through which we have analyzed the best number of local iterations at the devices. Furthermore, we have added Figure 2 illustrating the empirical variance, as well as the peak-to-average ratio of the vector to be quantized at the PS with different schemes.

---

### Decision · Program_Chairs · 2021-01-07
**Final Decision**

**Decision:**

Reject

**Comment:**

Although the reviewers acknowledge some contributions of the paper, it has some limitations on both theoretical results and numerical experiments. It is still unclear about the effectiveness of the proposed method. The authors should consider the following issues for the future submission:

1) The justification of $\tau$ is not clear in federated learning with respect to communication efficiency (please see Reviewer 1’s comments).

2) The bounded stochastic gradient assumption is not applicable in the strongly convex case. This issue has been discussed clearly in [Nguyen et. al, “SGD and Hogwild! Convergence Without the Bounded Gradients Assumption”, ICML 2018]. Therefore, the constant G in Section 3.2. would damage the complexity bound since it could be arbitrarily large.

3) Although the goal is to illustrate the benefits of the proposed quantization approach, the numerical experiments and the theoretical contributions are limited. The theoretical results are incremental from the existing optimization theory (both strongly convex and non-convex). Moreover, network architecture and data sets are not enough to justify the efficiency of the method.